# Neurons and neuronal activity control gene expression in astrocytes to regulate their development and metabolism

Philip Hasel[1], Owen Dando[1,2,3,*], Zoeb Jiwaji[1,2,*], Paul Baxter[1], Alison C. Todd[1], Samuel Heron[4], Nóra M. Márkus[1], Jamie McQueen[1], David W. Hampton[2], Megan Torvell[2], Sachin S. Tiwari[2], Sean McKay[1], Abel Eraso-Pichot[5], Antonio Zorzano[6,7,8], Roser Masgrau[5], Elena Galea[5,9], Siddharthan Chandran[2,3], David J.A. Wyllie[1], T. Ian Simpson[4] & Giles E. Hardingham[1,10]

The influence that neurons exert on astrocytic function is poorly understood. To investigate this, we first developed a system combining cortical neurons and astrocytes from closely related species, followed by RNA-seq and *in silico* species separation. This approach uncovers a wide programme of neuron-induced astrocytic gene expression, involving Notch signalling, which drives and maintains astrocytic maturity and neurotransmitter uptake function, is conserved in human development, and is disrupted by neurodegeneration. Separately, hundreds of astrocytic genes are acutely regulated by synaptic activity via mechanisms involving cAMP/PKA-dependent CREB activation. This includes the coordinated activity-dependent upregulation of major astrocytic components of the astrocyte–neuron lactate shuttle, leading to a CREB-dependent increase in astrocytic glucose metabolism and elevated lactate export. Moreover, the groups of astrocytic genes induced by neurons or neuronal activity both show age-dependent decline in humans. Thus, neurons and neuronal activity regulate the astrocytic transcriptome with the potential to shape astrocyte–neuron metabolic cooperation.

[1] Deanery of Biomedical Sciences, Edinburgh Medical School, University of Edinburgh, Edinburgh EH8 9XD, UK. [2] MRC Centre for Regenerative Medicine, University of Edinburgh, Edinburgh EH16 4SB, UK. [3] Centre for Brain Development and Repair, Institute for Stem Cell Biology and Regenerative Medicine, National Centre for Biological Sciences, Bangalore 560065, India. [4] School of Informatics, University of Edinburgh, Edinburgh EH8 9AB, UK. [5] Institut de Neurociències and Departament de Bioquímica i Biologia Molecular, Unitat de Bioquímica de Medicina, Edifici M, Universitat Autònoma de Barcelona, Bellaterra, Barcelona 08193, Spain. [6] Institute for Research in Biomedicine, Barcelona 08028, Spain. [7] Department of Biochemistry and Molecular Biology, University of Barcelona, Barcelona 08028, Spain. [8] Centro de Investigación Biomédica en Red de Diabetes y Enfermedades Metabólicas Asociadas (CIBERDEM), Instituto de Salud Carlos III (ISCIII), Madrid 28029, Spain. [9] Institució Catalana De Recerca I Estudis Avançats (ICREA), Passeig Lluís Companys 23, Barcelona, Catalonia, 08010, Spain. [10] UK Dementia Research Institute at The University of Edinburgh, Edinburgh Medical School, 47 Little France Crescent, Edinburgh EH16 4TJ, UK. * These authors contributed equally to this work. Correspondence and requests for materials should be addressed to G.E.H. (email: Giles.Hardingham@ed.ac.uk).

Signalling between neurons triggers programmes of gene expression that mediate specific functions in development and maturity. Signalling between neurons and glia is also implicated in diverse processes, however, insight into the scope and mechanism by which different cell types in the neuro-glial unit influence each other is lacking. A notable example is neuron-to-astrocyte signalling: neurons induce the classic stellate morphology in astrocytes, resembling their appearance *in vivo*[1]. However, knowledge of the transcriptional changes that accompany this transformation are restricted to three astrocytic genes (*Slc1a2*, *Slc1a3* and *Gja1* (refs 2,3)) induced by neurons. Similarly, despite astrocytes playing a critical role in the uptake of synaptically released neurotransmitters, whether synaptic activity controls astrocytic gene expression is unclear. This contrasts with the wealth of knowledge concerning the mechanisms and the roles of activity-dependent gene expression in neurons[4–7].

Understanding non-cell-autonomously regulated gene expression poses challenges associated with the physical separation of cell types before transcriptome analysis by methods including RNA-seq. Immunopanning or FACS are powerful approaches, but are subject to off-target cell type contamination, and cause aberrant induction of transcriptional stress responses, or the loss of material from delicate subcellular regions[8–10]. Theoretically, if RNA-seq reads could be unambiguously attributed to one cell type or another, physical sorting could be avoided. We investigated the possibility of achieving this by deriving the distinct cell types from different species[11]. We chose rat and mouse, two species that we reasoned may be closely enough related to enable questions of non-cell-autonomous signalling to be answered. We used this system as a starting point to investigate how and to what extent neurons and neuronal activity control astrocytic gene expression, as well as the underlying mechanisms and functional consequences.

## Results

**Unsorting rat and mouse RNA-seq reads *in silico*.** We first established the feasibility of our species-specific sorting (SSS) RNA-seq workflow. In the vast majority of genes, the majority of mouse RNA-seq paired-end reads can be unambiguously attributed to mouse (as opposed to rat), using both a simulated set of reads (Supplementary Fig. 1a; Supplementary Data 1) as well as a 'real-world' RNA-seq data set of mouse cortical neurons (Supplementary Fig. 1b; Supplementary Data 2). Also, compared to normal read mapping, the SSS workflow does not substantially affect differential gene expression analysis (DGE) when performed on a single-species (mouse) RNA-seq data-set taken from neurons treated ± bicuculline + 4-aminopyridine (BiC/4-AP) to induce synaptic activity via network disinhibition[12]: we observed a tight correlation in fold-induction, comparing normal and SSS approaches (Supplementary Fig. 1c; Supplementary Data 3). There was also little change in the adjusted *P* values: of the 4,632 genes altered by BiC/4-AP treatment, only 33 were rendered insignificant by the SSS approach (Supplementary Fig. 1d; Supplementary Data 4). Large *P* value deviations were more likely to occur in the small number of genes that had lost a lot of reads due to the SSS approach. However, the tiny number of genes affected leads us to conclude that the SSS workflow does not substantially impact on DGE analysis.

**Neurons transform astrocytic transcriptome and function.** To profile the influence of neurons on the astrocytic transcriptome, we cultured primary mouse cortical astrocytes, onto which we co-cultured primary rat cortical neurons. The mouse cortical astrocytes are >96% GFAP positive[13], which we confirmed with

another marker (Aldh1l1, 99.4 ± 0.2% positive ($n = 4$), Supplementary Fig. 2a). The astrocytes were <0.1% Neuro-Chrom$^+$ neurons and <0.1% Iba$^+$ microglia.

Compared to parallel astrocytic mono-cultures, co-cultured astrocytes developed the expected stellate morphology over 9 days (Fig. 1a,b). At this point, RNA was extracted (3 biological replicates), RNA-seq performed and SSS employed to identify the mouse (that is, astrocyte) reads. DGE analysis revealed widespread up- and down-regulation of astrocytic gene expression due to the presence of neurons (Fig. 1c; Supplementary Data 5), including previously identified genes (*Slc1a2*, *Slc1a3* and *Gja1* (refs 2,3)). Four hundred and ten genes were induced more than twofold and 353 genes repressed more than twofold (Adjusted *P* value <0.05, Supplementary Data 5).

A notable cluster of upregulated genes is involved in the uptake and metabolism of neurotransmitters, including GABA (*Slc6a1*, *Slc6a11* and *Abat*), NAAG (*Slc17a5*), biogenic amine neurotransmitters (*Slc29a2* and *Maob*) and glutamate (*Slc1a2*, *Slc1a3*, *Glul* and *Glud1*; Supplementary Data 6). Focussing on glutamate uptake, we confirmed that these transcriptional changes are reflected at the functional level, by measuring the electrogenic currents of the glutamate transporters EAAT1/GLAST (*Slc1a3*), and EAAT2/GLT-1 (*Slc1a2*), using whole-cell voltage clamp recordings. Astrocytic EAAT currents[14] were increased around 10-fold by neuronal co-culture (Fig. 1d,e), while astrocytic resting membrane potential and passive membrane conductance were unaffected (Supplementary Fig. 2b,c)[15,16]. In addition, many cytoskeletal and extracellular matrix genes were altered (Supplementary Data 5), consistent with the profound changes in astrocytic morphology. Astrocyte-specific marker and endfoot component *Aqp4* was also induced, but other markers, *S100b*, *Aldh1l1* and *Gfap* were unaffected (Fig. 1c). Thus, neuron-induced changes to the morphology of astrocytes are associated with marked changes to their transcriptome and functional properties.

To assess the advantages of this approach over imperfect physical separation approaches, we firstly simulated a sorting process that achieved 95% purity, by 'contaminating' the mouse astrocytic mRNA with mouse cortical neuronal mRNA at a ratio of 95:5, by number of cells collected, and performed RNA-seq. The effects were substantial: 863 genes were expressed more than two-fold higher in the 95% sample, compared to the pure astrocytic sample (Supplementary Fig. 3a, Supplementary Data 7), of which 216 were expressed >10-fold higher. Second, we investigated the induction of transcriptional stress responses to physical sorting methods[9]. Both simple trypsinization, immunopanning and FACS protocols applied to a homogeneous population of cultured astrocytes induced multiple immediate early genes (Supplementary Fig. 3b–d). These data show key advantages of *in silico* read separation in avoiding DGE artefacts introduced by physical sorting.

**Neurons induce an *in vivo*-like transcriptome in astrocytes.** While neuronal co-culture promotes a more *in vivo*-like morphology and functional profile in astrocytes, we wanted to determine whether the observed changes in the astrocytic transcriptome were consistent with a more *in vivo*-like shift. We compared our data to a microarray study that characterized differences in gene expression between mono-cultures of astrocytes *in vitro*, and astrocytes acutely sorted from the cortex *in vivo*[17]. The group of 695 genes elevated *in vivo* more than twofold and meeting expression level cut-off criteria (see Methods) in the data of Cahoy *et al.* were significantly induced by neuronal co-culture, compared to mono-culture in our data set (Fig. 2a; Supplementary Data 8). The group of 654 genes

repressed *in vivo* more than twofold in the data set of Cahoy *et al.* were significantly repressed by neuronal co-culture in our data set (Fig. 2b; Supplementary Data 9). These analyses suggest that the mature *in vivo* gene expression profile of astrocytes may be at least in part due to neuron–astrocyte communication, since it can be mimicked *in vitro* by neuronal co-culture.

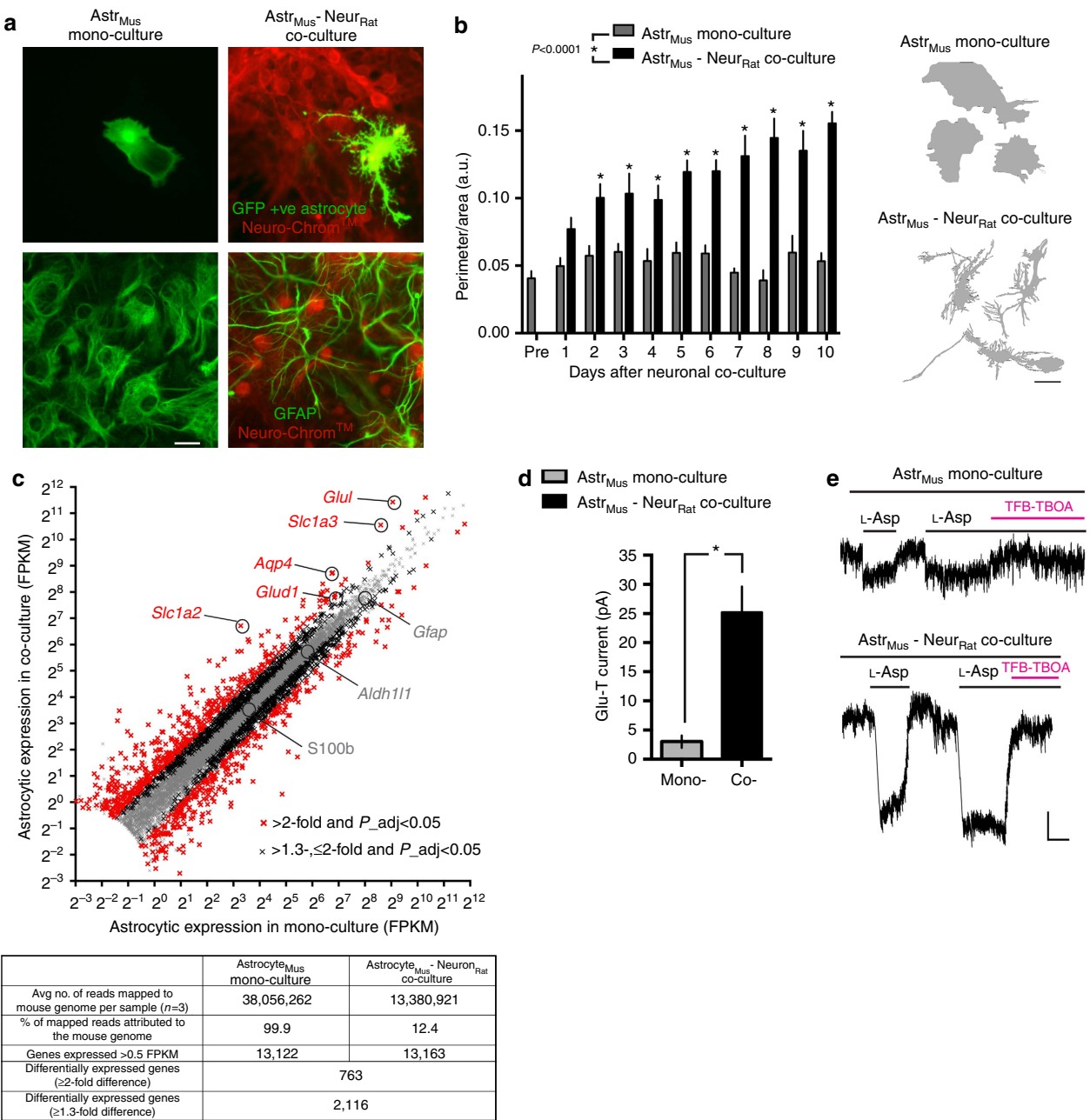

**Figure 1 | Mixed-species RNA-seq uncovers neuronally induced astrocytic gene expression. (a)** The mixed-species co-culture system recapitulates characteristic neuron-induced changes to astrocyte morphology. Upper: example picture of a GFP-transfected mouse cortical astrocyte cultured for 10 days with (right) or without (left) rat cortical neurons. Neurons are identified by immunofluorescent staining with pan-neuronal antibody cocktail Neuro-Chrom. Lower: Staining of mouse astrocyte monoculture (left) and mixed-species co-culture (right) for astrocyte-specific gene Gfap (plus Neuro-Chrom). Scale bar, 20 µm. **(b)** Perimeter:area ratio of GFP + astrocytes described in **a** at the indicated times (days in vitro). 6–16 cells were analysed for each culture type, for each day. *$P < 0.05$ compared to astrocytic monoculture on that day (two-way analysis of variance (ANOVA), plus Holm-Sidak's *post hoc* test). Example outlines are shown on the right. Scale bar, 20 µm. **(c)** Neurons promote widespread gene expression changes in astrocytes. RNA from mixed-species mouse astrocyte/rat neuron co-cultures ($n = 3$ biological replicates) was subjected to RNA-seq, followed by SSS workflow to identify reads that were unambiguously mouse (that is, astrocytic) in origin. The same workflow was applied to mouse astrocytic mono-cultures. Expression of genes (FPKM) in astrocytes in the presence or absence of neurons is plotted for all genes expressed > 0.5 FPKM average across mono- and co-cultures. Red and black crosses indicate the astrocytic genes induced or repressed by neurons more than twofold or > 1.3, ≤ 2-fold respectively (DESeq2 $P$\_adj < 0.05). See Supplementary Data 5. The table summarizes read depth, species sorting and number of differentially expressed genes. **(d)** Neurons promote increased glutamate transporter capacity in astrocytes. Mouse astrocytes cultured with or without neurons (as in **a–c**) were subject to whole-cell voltage-clamp recording of glutamate transporter currents, measured upon application of 200 µM aspartate (see Methods). Currents were inhibited by the glutamate transporter blocker TFB-TBOA (20 µM). *$P < 0.05$ (unpaired *t*-test, $n = 13$ (mono), 22 (co-culture)). **(e)** Example traces. Scale bar, 5 s, 10 pA. All error bars represent s.e.m.

**Neuron-induced changes to human astrocytes.** To determine the extent to which the transcriptional programmes induced by neurons are relevant to human astrocytes, we created co-cultures in which rat neurons were overlaid onto primary human fetal astrocytes. The presence of neurons transformed human astrocytes from a simple polygonal shape to a complex

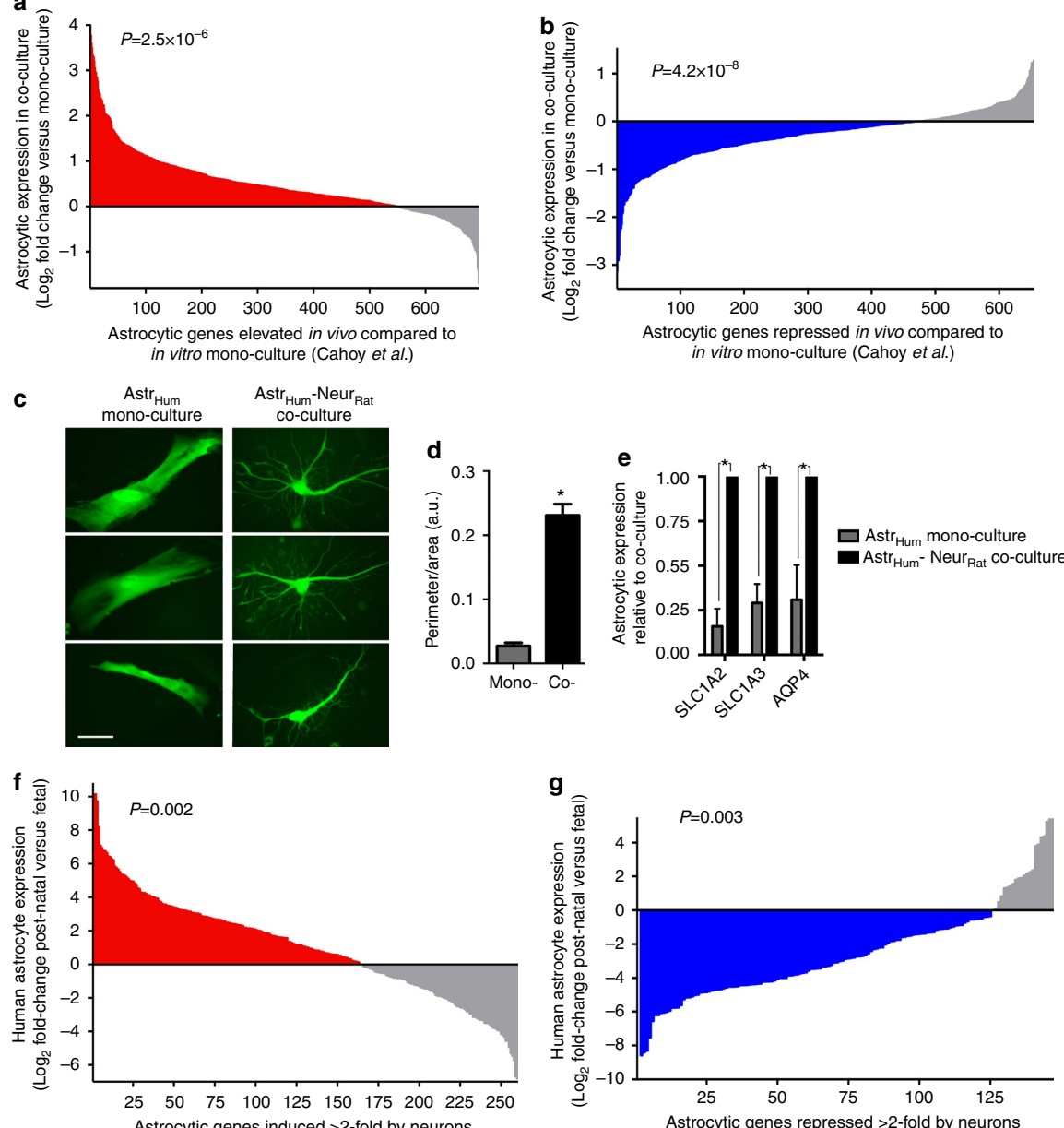

**Figure 2 | Linking neuron-induced astrocytic gene expression to mouse and human development. (a,b)** Neurons are sufficient to drive the astrocytic transcriptome towards an *in vivo* profile. Gene fold-change caused by neuronal co-culture (as analysed in Fig. 1c) is shown for those genes identified by Cahoy *et al.*[17] in a microarray screen as being either elevated (**a**) or lowered (**b**) in astrocytes *in vivo*, compared to *in vitro* mono-culture[17]. Genes expressed > 0.5 FKPM in either mono- or co-culture are shown. In some cases gene names are updated from those quoted in Cahoy *et al.*[17]. Within the group of genes elevated *in vivo* (**a**, 695 genes induced more than twofold) the influence of neurons on astrocytic expression is shown, ranked according to fold-change. $P = 2.5E - 6$ (paired *t*-test comparing FPKM of these 695 genes in mono-culture versus co-culture). Within the group of genes repressed *in vivo* (**b**, 654 genes repressed more than twofold) the influence of neurons on astrocytic expression is shown, ranked according to fold-change. $P = 4.2E - 8$ (paired *t*-test comparing FPKM of these 654 genes in mono-culture versus co-culture). All the data are available in Supplementary Data 8 and 9. (**c,d**) Primary human astrocytes are morphologically transformed by co-culture with neurons. Experiment performed as per Fig. 1a,b except that instead of mouse astrocytes, primary human fetal astrocytes were used. *$P < 0.05$ (unpaired *t*-test, $n = 6$ (mono-), 6 (co-culture)). Scale bar, 50 µm. (**e**) Human astrocytic gene expression in mono- and co-culture measured using species-specific qPCR primers. *$P < 0.05$ (unpaired *t*-test, $n = 5$). (**f,g**) The neuronal influence on expression of astrocytic genes aligns significantly with the developmental trajectory of those genes in the human brain. Astrocytic genes induced more than twofold (**f**) or repressed more than twofold (**g**) (Fig. 1c) were cross-referenced to the data in Zhang *et al.*[18], and the fold change in expression in human astrocytes (post-natal versus fetal) from this study calculated. Only genes meeting an expression threshold of > 0.5 FPKM in both studies are shown. (**f**) *$P = 0.002$ (paired *t*-test comparing average FPKM of these genes in post-natal versus fetal). (**g**) *$P = 0.003$ (paired *t*-test comparing FPKM of these genes in post-natal versus fetal). Data are available in Supplementary Data 17 and 18. All error bars represent s.e.m.

stellate morphology (Fig. 2c,d). Moreover, neuron-induced increases in astrocytic *SLC1A2*, *SLC1A3* and *AQP4* expression were observed (Fig. 2e), suggesting conservation of neuron-to-astrocyte signalling, and that human astrocytes also receive maturation cues from neurons.

We next investigated the extent to which the astrocytic genes we identified as being neuronally regulated (NR) are differentially expressed in astrocytes in human development. We compared our data to a recent RNA-seq analysis of fetal and post-natal astrocytes extracted from human tissue from individuals of a variety of ages[18]. The group of genes we identified as being induced more than twofold in astrocytes by neurons was significantly upregulated in post-natal astrocytes (age 8–63 years) compared to fetal astrocytes (18–19 weeks gestation), (Fig. 2f, $P = 0.002$ paired $t$-test, Supplementary Data 17). In a similar vein, the group of genes we identified as being repressed more than twofold in astrocytes by neurons was significantly downregulated in post-natal astrocytes (age 8–63 years) compared to fetal astrocytes (18–19 weeks gestation; Fig. 2g; $P = 0.003$, paired $t$-test; Supplementary Data 18). Thus, the neuronal influence on expression of astrocytic genes (up or down) aligns significantly with the developmental trajectory of those genes in the human brain.

**Neuron-induced astrocytic genes are deregulated *ex vivo*.** While neuron-derived signals are sufficient to drive the astrocytic transcriptome towards a more *in vivo*-like profile in co-culture (Fig. 2a,b), we wanted to determine whether similar signals are responsible for maintaining this profile *in vivo*. We investigated whether astrocytic expression of neuronally induced genes declines if astrocytes are removed from their normal *in vivo* environment. Astrocytes were isolated from the mouse cortex by immunopanning at P7, when the astrocytic transcriptome is near maturity[19,20]. We measured expression of a panel of neuron-induced astrocytic genes both immediately post-isolation and again after 4 days of *ex vivo* maintenance in isolation from neurons, and observed a marked decline in expression (Fig. 3a). Moreover, subsequent addition of neurons reversed this loss of expression (Fig. 3b), suggesting that *in vivo*, a neuronally derived signal is responsible for maintaining key aspects of the astrocytic transcriptome.

As an additional control to rule out that any astrocytic gene induction in our co-culture system was due to species differences of the neurons and astrocytes, we analysed expression of the NR astrocytic genes shown in Fig. 3a,b whose induction could be tracked in a single species co-culture by virtue of their expression being >10-fold lower in neurons than astrocytes (*Hes5*, *Dio2*, *Slco1c1*, *Glul* and *Cldn10*). This group of genes studied was indeed significantly elevated in the single species co-culture, compared to astrocytic mono-culture (Supplementary Fig. 4a). We also confirmed that the functional impact of neurons in inducing mouse astrocytic glutamate transporter capacity is observed in a single species mouse neuron/astrocyte co-culture (Supplementary Fig. 4b) consistent with our observations based on the mixed-species preparation (Fig. 1d).

**Neuron-induced astrocytic genes are repressed by tauopathy.** The reversibility of neuron-induced influences on astrocytic gene expression raises the possibility that neurodegeneration could lead to a decline in NR astrocytic gene expression. We employed the Thy1-P301S neurodegenerative tauopathy model[21,22], focussing on the spinal cord, where neurodegeneration is most abundant[22], and confirmed the presence of phosphorylated tau (Fig. 3c) and neurodegeneration (Fig. 3d). We studied the expression of those genes we identified as being strongly up-regulated by neurons ($\geq 5$-fold) whose expression in a mixed cell population could be reasonably attributed to astrocytes (astrocytic expression $\geq 10$-fold higher than in neurons, oligodendrocytes, microglia or endothelial cells, based on the data within Zhang *et al.*[23], nine genes in total). We also confirmed that as a group they were significantly induced by neurons in spinal astrocytes (Supplementary Fig. 4c) and that spinal astrocytes underwent a similar neuron-induced morphological transformation (Supplementary Fig. 4d).

The expression of these genes in wild type versus P301S mice was normalized to astrocytic marker *Aldh1l1* (refs 17,18) whose expression is not influenced by neuronal contact (Fig. 1c). *Aldh1l1* also has stable expression *in vivo* following inflammatory or ischaemic injury[24], and is unaltered in the P301S versus wild-type spinal cord at the mRNA or protein level (Supplementary Fig. 4e,f,g). The group of nine genes was significantly downregulated in the 20-week-old P301S mouse (Fig. 3e; $P < 0.0001$, two-way analysis of variance). Thus, neurodegeneration triggered by a neuron-specific transgene leads to a decline in the astrocytic expression of NR genes *in vivo*.

**Notch contributes to neuron-induced astrocytic maturation.** We next investigated what neuron-derived signal(s) may be responsible for the induction and maintenance of these genes in astrocytes, since signals involved in the developmental maturation of astrocytes are not well understood[1].

We noted that expression of Notch target genes *Hes5* and *Hey2* were (i) induced by rat neurons in mouse astrocytes (Supplementary Data 5), (ii) repressed when astrocytes were removed from their *in vivo* environment (Fig. 3a) and (iii) induced again when neurons were overlaid onto *ex vivo* astrocytes (Fig. 3b). *Hes5* was also downregulated *in vivo* in the P301S mouse (Fig. 3e) and induced in mouse astrocytes by mouse neurons (Supplementary Fig. 4a). These observations suggested to us a potential role for neuron-derived Notch signalling. Analysis of expression levels of Notch and Notch ligands in neurons and astrocytes revealed that neurons (but not astrocytes) strongly express Notch ligands, particularly *Jag2* and *Dlk2*, while astrocytes strongly express receptors *Notch1* and *Notch2* (Fig. 4a). To investigate neuron-to-astrocyte Notch signalling, astrocytes were transfected with a Notch reporter (CBF1-luciferase) containing binding sites for the transcription factor CBF1, which is converted from repressor to activator upon binding the intracellular domain of Notch[25]. Astrocytic CBF1-luciferase activity was strongly induced in the presence of neurons, compared to mono-culture (Fig. 4b). Moreover, the neuron-dependent induction of CBF1-luciferase activity (and of *Hes5* and *Hey2*) was blocked by inhibiting endogenous Notch signalling by treatment with the Notch pathway inhibitor DAPT, which prevents cleavage of Notch by γ-secretase (Fig. 4b,c). The neuron-dependent induction of a panel of other neuronally induced astrocytic genes was also reduced or blocked by DAPT, including the glutamate transporter *Slc1a2* (Fig. 4c). DAPT treatment was also found to inhibit the neuron-induced increase in astrocytic glutamate transporter EAAT activity (Fig. 4e,f). Interestingly, DAPT did not influence the neuron-induced transformation of astrocytic morphology (Supplementary Fig. 5a), suggesting that neuronal-induced functional and morphological changes can be uncoupled and are mediated by different mechanisms. Finally, to test whether canonical Notch signalling is sufficient to boost astrocytic glutamate uptake capacity, we expressed a constitutively active form of CBF1 (CBF1-VP16 (ref. 26; Fig. 4g)) in neuron-free astrocyte mono-cultures, and found that this did indeed increase glutamate transporter currents (Fig. 4h). These experiments show that

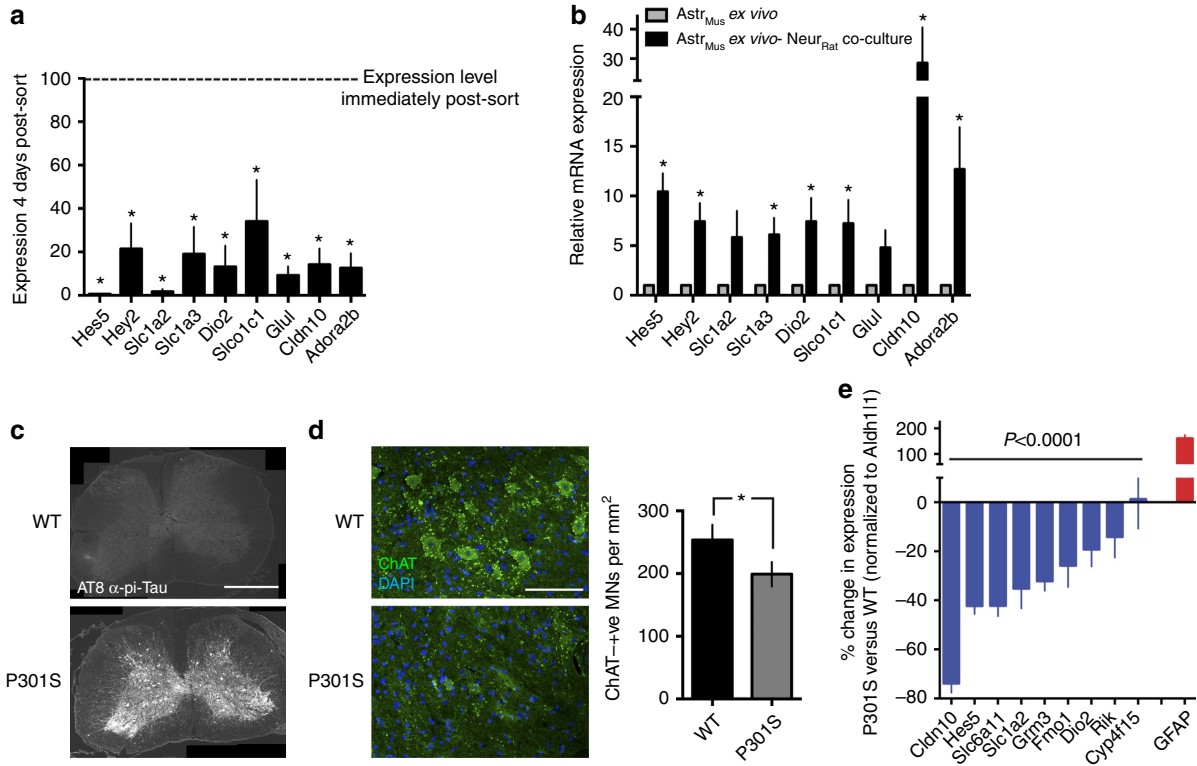

**Figure 3 | Neuron-induced astrocytic gene expression is reversible and disrupted in neurodegenerative disease.** (**a**) Mouse astrocytes were isolated by immunopanning (using an anti-GLAST antibody) at P7. Expression of the indicated neuron-induced astrocytic genes was measured both immediately post-isolation and after 4 days of *ex vivo* maintenance (neuron-free). *$P < 0.05$ unpaired *t*-test, ($n = 5$). (**b**) Mouse astrocytes isolated as in **a**, and maintained for 22 days in culture, with (black) or without (grey) the last 11 days being in the presence of rat neurons. *$P < 0.05$ unpaired *t*-test, ($n = 7$). (**c**) Example micrographs showing phospho-tau immunoreactivity in spinal cord sections of the Thy1-P301S transgenic mouse (at week 20). Scale bar, 500 μm. (**d**) (Left) Example micrographs showing loss of ChAT-positive motor neurons in spinal cord sections of the Thy1-P301S transgenic mouse at week 20. Scale bar, 50 μm. (Right) Quantitation of loss of ChAT-positive motor neurons in spinal cord sections of the Thy1-P301S transgenic mouse. *$P < 0.05$ unpaired *t*-test, ($n = 5$ animals of both genotypes). (**e**) Expression of the indicated astrocyte-specific, neuron-induced genes in the spinal cord of week 20 wild type (WT) and P301S mice, normalized to astrocyte-specific gene *Aldh1l1*. Rik: *2900052N01Rik*. $P < 0.0001$, two-way analysis of variance, $P$ value indicates main effect of genotype ($n = 6$ animals of both genotypes). All error bars represent s.e.m.

Notch signalling is necessary and sufficient to drive glutamate uptake capacity in astrocytes.

We hypothesized that many of the genes whose induction is repressed by DAPT are likely to be a secondary response to Notch signalling, as opposed to direct Notch targets. We expressed a strong activator of Notch signalling (intracellular domain of Notch1: Notch1-IC) in mono-cultured astrocytes and analysed gene expression 48 h post-transfection. We observed that, while *Hes5* and *Hey2* were induced, the other genes were not (Supplementary Fig. 5b), suggesting that they may be a delayed downstream consequence of Notch signalling.

In addition, we observed that rat neurons can also induce *Hes5* and *Hey2* in human astrocytes (Fig. 4d), in addition to inducing the glutamate transporters *Slc1a2* and *Slc1a3* (Fig. 2e), suggesting that the ability of astrocytes to receive neuron-derived Notch-dependent maturation signals is evolutionarily conserved. Moreover, we confirmed that this neuronally-derived signal is not sensitive to the neuronal species employed. We generated 'reversed' mixed-species co-cultures comprised of rat cortical astrocytes in the presence of mouse cortical neurons, and observed a similar neuron-induced astrocytic expression of *Hes5*, *Hey2*, *Slc1a2* and *Slc1a3* (Supplementary Fig. 5c) that we observed with rat neurons co-cultured with either mouse or human astrocytes. This strong inter-species cross-compatibility is consistent with the high amino acid conservation between

rodents and humans for both Notch and Notch ligand interaction domains ($> 95\%$).

Another issue we addressed is the possibility that small numbers of non-neuronal cells in the rat neuronal preparation were important for inducing the changes observed in mouse astrocytic gene expression. In the co-culture system, the rat neuronal preparation is co-cultured in zero serum to limit proliferation of non-neuronal cells. To limit this further, we performed an additional experiment with an adapted protocol such that the rat neuronal preparation is added to the astrocytes in the continuous presence of the anti-mitotic agent AraC. Under these conditions, $> 99\%$ of the rat neuronal preparation is Neuro-Chrom$^+$ neurons, $< 0.01\%$ Aldh1l1$^+$ astrocytes, $< 0.01\%$ Iba1$^+$ microglia, after 9 days in culture (Supplementary Fig. 5d). The only caveat to these experiments is that the astrocytes do not appear as healthy in AraC and there is a small level of cell death. Nevertheless, the group of putative Notch-dependent NR astrocytic genes studied in Fig. 4c are still induced (Supplementary Fig. 5e), consistent with the notion that neurons are the key mediators of these changes, although a minor contribution of other cell types cannot be completely ruled out. Collectively, these data point to neuron-to-astrocyte juxtacrine Notch signalling as an important inducer and maintainer of astrocytic functional maturation, in addition to its known role in astrocytic fate specification[1,27].

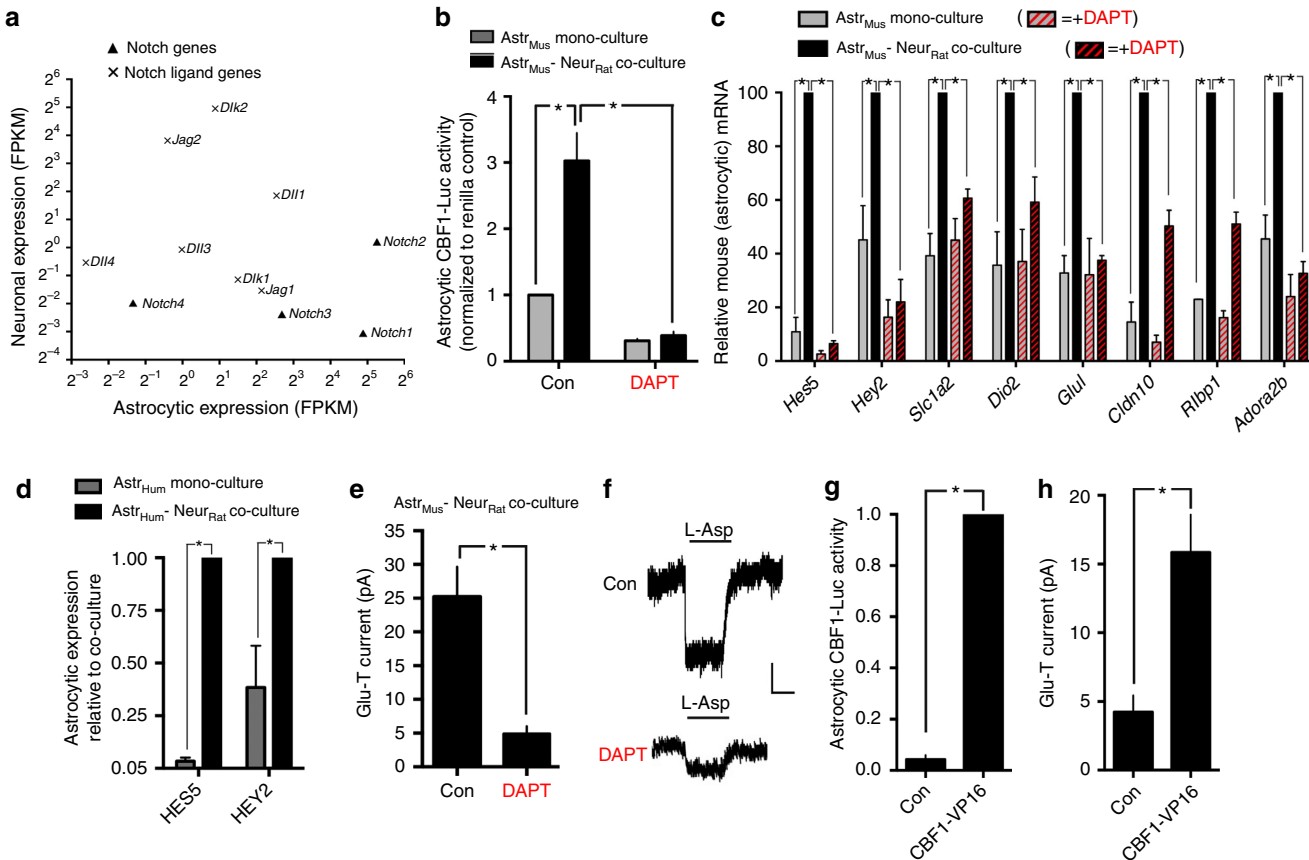

**Figure 4 | Notch signalling contributes to neuron-induced astrocytic gene expression and functional maturation.** (**a**) Mean expression levels ($n = 3$) of Notch and Notch ligand genes are compared between pure mouse astrocyte cultures and pure mouse neuronal cultures, derived from RNA-seq data and calculated as FPKM. (**b**) Neurons induce astrocytic Notch/CBF1-dependent gene expression. Mouse astrocytes were transfected with a CBF1-luciferase reporter (plus renilla control) before rat neurons were overlaid (or not) for 9 days in the presence or absence of the γ-secretase inhibitor DAPT (10 μM, Tocris). Firefly luciferase activity was normalized to Renilla transfection control. *$P < 0.05$ (two-way analysis of variance (ANOVA) plus Sidak's *post hoc* test ($n = 3$)). (**c**) Mouse astrocyte mono-cultures and mouse astrocyte/rat-neuron co-cultures were treated with DAPT, and gene expression analysed at DIV9, normalized to housekeeping gene *H1f0*, using mouse species-specific PCR primers. *$P < 0.05$ (two-way ANOVA, plus Holm-Sidak's multiple comparisons test, $n = 3$). (**d**) Neuron-induced expression of Notch target genes analysed in human primary astrocytes. *$P < 0.05$ (unpaired t-test, $n = 5$). (**e,f**) DAPT treatment inhibits the neuron-induced increase in glutamate transporter capacity in astrocytes in response to 200 μM L-Aspartate. *$P < 0.05$ (unpaired t-test, $n = 22$ Con, $n = 8$ DAPT). (**f**) Example traces. Scale bar, 5 s, 10 pA. (**g**) Verification that CBF1-VP16 induces Notch/CBF1-dependent gene expression, compared to control plasmid (β-globin). Neuron-free astrocyte mono-cultures were used for this experiment. *$P < 0.05$ (t-test). (**h**) CBF1-dependent gene expression is sufficient to induce glutamate transporter currents in astrocytes. Astrocyte mono-cultures were transfected as indicated (plus eGFP marker) and glutamate transporter currents in response to 200 μM L-Aspartate application measured 9 days later. *$P < 0.05$ (unpaired t-test, $n = 8$ Con, $n = 10$ CBF1-VP16). All error bars represent s.e.m.

**Synaptic activity directs astrocytic gene expression**. We next determined whether astrocytic gene expression is also subject to acute control by synaptic activity. To alter neuronal activity within the mixed-species co-culture system we incubated the cells overnight in the voltage-gated Na$^+$ channel blocker tetrodotoxin (TTX), to block action potential (AP) firing. We then washed out the TTX in the presence of the GABA$_A$ (γ-aminobutyric acid) receptor antagonist bicuculline (BiC), to trigger excitatory synaptic activity associated with bursts of synchronous action potential firing (Fig. 5a). We performed RNA-seq on RNA extracted 16 h post-washout, using the SSS workflow to identify mouse (astrocytic) reads, and compared these to a control co-culture treated the same way except that TTX remained in the medium throughout. DGE analysis revealed that 351 astrocytic genes were significantly induced >1.3-fold by neuronal TTX-sensitive synaptic activity (57 genes more than twofold), while 154 genes were repressed >1.3-fold (4 genes more than two-fold; Fig. 5b; Supplementary Data 10). We

confirmed that BiC did not induce astrocytic gene expression in the absence of neuronal activity (Fig. 5c). Since this network activity is glutamatergic in nature, we tested the influence of prolonging synaptic glutamate levels with the glutamate uptake inhibitor TBOA. Indeed, TBOA resulted in slightly prolonged AP bursts (Supplementary Fig. 6a), prolonged exposure of astrocytes to glutamate (assayed using iGluSnFR, Fig. 5d) and triggered stronger activity-dependent induction of astrocytic genes (Fig. 5e,f; Supplementary Data 11). Thus, using mixed-species RNA-seq we have identified a novel type of synapse-to-nucleus signalling; one that targets astrocytic genes rather than neuronal ones. We refer to these genes as astrocytic activity-response genes (AAR genes), to distinguish them from the NR genes identified in Fig. 1c. Of note, it is unlikely that glutamate itself is the key messenger sensed by astrocytes resulting in signal transduction to *de novo* gene expression: exogenous glutamate application was unable to induce a panel of AAR genes (Supplementary Fig. 6b). In contrast, exogenous application of ATP, a molecule whose

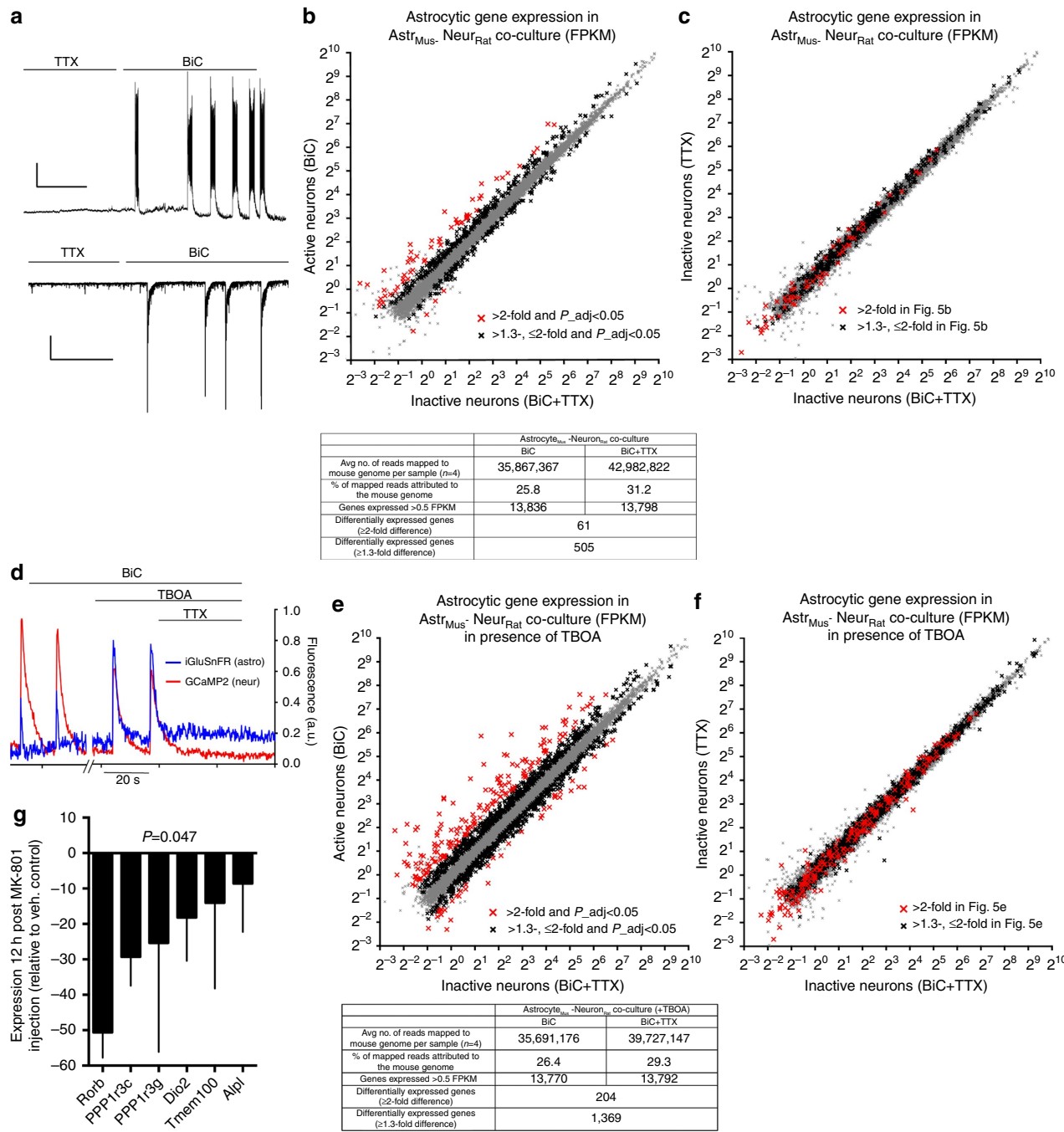

**Figure 5 | Synaptic activity regulates astrocytic gene expression.** (**a**) Mixed-species astrocyte–neuron co-cultures were transferred into TTX-containing medium for 22 h, after which it was washed out in the presence of BiC. Whole-cell current-clamp (upper) and voltage-clamp (lower) recording of the resultant burst activity. Scale bar, 15 mV, 30 s (upper); 50 pA, 30 s (lower). (**b**) Mixed-species co-cultures were treated as in **a**, or a control that was 'washed' but remained in TTX (BiC + TTX), and RNA extracted at 16 h post-wash. SSS RNA-seq identified mouse (that is, astrocyte) reads. Expression of genes (FPKM) in astrocytes ± neuronal synaptic activity is plotted for the genes expressed > 0.5 FPKM averaged over the conditions. Red and black crosses indicate the astrocytic genes changed by neurons more than twofold or > 1.3, ≤2-fold respectively (DESeq2 $P$_adj < 0.05, $n = 4$ biological replicates). See Supplementary Data 10. (**c**) To confirm that BiC was not having a direct effect on astrocytic gene expression independent of AP firing, gene expression in the (BiC + TTX) condition was compared to a condition where the neurons remained in TTX. The crosses indicate those genes significantly changed in **b**. (**d**) Example trace depicting an astrocyte–neuron co-culture in which astrocytes had been transfected with iGluSnFR[68] before neuronal co-culture, after which neurons were transfected with GCaMP2. GCaMP2-positive neurons and iGluSnFR-positive astrocytes were imaged concurrently in the presence where indicated of BiC (50 μM), TBOA (50 μM) and TTX. (**e**) Impairing glutamate re-uptake enhances activity-dependent astrocytic gene regulation. Experiment performed exactly as in **b** except co-cultures were treated with TBOA (50 μM) at the time of BiC addition ($n = 4$). See Supplementary Data 11. (**f**) To confirm that BiC + TBOA were not having a direct effect on astrocytic gene expression, expression in the (BiC + TBOA + TTX) condition was compared to a further condition where the neurons remained in TTX. The crosses indicate those genes significantly changed in **e**. (**g**) P7 mice were treated with MK-801 (0.5 mg kg$^{-1}$) or PBS and cortical expression of the indicated astrocyte-specific, activity-regulated genes analysed 12 h post-injection. *$P < 0.05$ (2-way analysis of variance, Con versus MK-801; $n = 8$ con, 8 MK), $P$ value indicates the main effect of drug treatment versus PBS. Error bars represent s.e.m.

metabolite adenosine can induce the *de novo* astrocytic gene expression (of *Ppp1r3c* (PTG)) in astrocytes via A2B receptors[28] was able to induce some of these genes (Supplementary Fig. 6b).

To further validate the mixed-species approach we wanted to rule out the possibility that co-culturing neurons and astrocytes of different species results in erroneous effects of synaptic activity on astrocytic gene expression. We focused on the genes whose activity-dependent induction we could track in astrocytes in a single-species co-culture, that is, genes highly enriched in astrocytes over neurons. Within the 56 genes induced more than twofold by BiC in astrocytes, we selected the 15 genes that fulfilled this enrichment criterion (expressed >10-fold higher in a mixed neuron–astrocyte culture compared to neurons alone, Supplementary Data 12) and studied their regulation in a single species (mouse) neuron–astrocyte co-culture treated ± BiC/4-AP to increase synaptic activity levels. All 15 genes were significantly induced by BiC/4-AP treatment, while their expression in an astrocyte-free neuronal mono-culture treated ± BiC/4-AP was extremely low (Supplementary Fig. 6d), strongly suggestive of an astrocytic locus for their induction in the co-cultures. In contrast, neuronally expressed genes were induced in neuronal mono-culture as well as neuron–astrocyte co-culture (Supplementary Fig. 6e). Thus, with this subset of genes, there is no evidence that having a different neuronal species results in erroneous effects of synaptic activity on astrocytic gene expression.

We next investigated whether neuronal activity can control astrocytic gene expression *in vivo*. We treated P7 mice with a mildly sedating dose of the NMDA (*N*-methyl-D-aspartate) receptor antagonist MK-801 (refs 12,29), and analysed expression of genes whose expression could be attributed to astrocytes. We looked at those genes (six) induced more than twofold by BiC that are expressed >10-fold higher in astrocytes than neurons, microglia or oligodendrocytes, according to a recent study[23], and robustly expressed in the cortical samples from control animals (at least 10% of the *Aldh1l1* level). MK-801 administration significantly repressed expression of these genes (Fig. 5g). Note that none of these genes are regulated directly by glutamate (in the presence or absence of MK-801; Supplementary Fig. 6b) meaning that MK-801 is unlikely to be having direct effects on astrocytic gene expression. Thus, neuronal activity may influence astrocytic gene expression *in vivo*.

**Synaptic activity induces astrocytic transcription via CREB.** Comparison of the induced NR and AAR genes revealed a modest overlap. Only 17 of the NR-genes induced more than twofold were also within the 56 genes induced more than twofold by BiC-induced neuronal activity, suggestive of distinct mechanisms of gene induction. We performed a promoter analysis of the genes induced more than twofold by both BiC and BiC + TBOA to look for enrichment in transcription factor consensus matrices. The highest enrichment was for members of the cAMP response element binding protein (CREB) family (Fig. 6a,b; Supplementary Data 13), a group of transcription factors responsive to both $Ca^{2+}$ and cAMP/PKA signalling, and important mediators of activity-dependent gene expression in neurons[4,6,7,30,31].

We performed simultaneous GCaMP2-based $Ca^{2+}$ imaging of neurons and astrocytes, which revealed an increase in astrocytic $Ca^{2+}$ transients upon BiC treatment, enhanced with TBOA application (Fig. 6c,d). Moreover, imaging astrocytic PKA activity with the FRET probe AKAR4 (ref. 32) also revealed an activity-dependent induction (Fig. 6e,f). Thus, second messenger pathways central to CREB activation are induced in astrocytes by synaptic activity. Moreover, synaptic activity induced astrocytic CREB-dependent gene expression (Fig. 6g, assayed using a CRE-luciferase reporter transfected into the astrocytes

before neuronal co-culture). This induction was inhibited by co-expression of the PKA inhibitor protein PKI (gene name *Pkia*)[33], implicating the cAMP–PKA pathway. We also observed that elevating cAMP levels by treatment with the adenylate cyclase activator forskolin was sufficient to induce a panel of AAR genes (*Ppp1r3c*, *Slco1c1*, *Dio2*, *Sod3* and *Scg2*), but (as expected) had no effect on members of the NR gene set, *Glul* and *Cldn10*. These observations were qualitatively similar in both co- (Fig. 6h) and mono-cultured astrocytes (Supplementary Fig. 6c). Thus, while neuronal co-culture is required for many functional changes to occur in astrocytes, the signalling machinery required to induce many AAR genes is intrinsic to astrocytes.

It was not possible to pharmacologically inhibit PKA to assess its role in activity-dependent astrocytic gene expression because the inhibitors tend to interfere with neuronal activity itself. Instead, we investigated the ability of portions of the promoters of the AAR genes *Scg2* and *Dio2* to confer activity inducibility on a luciferase reporter in astrocytes. Neuronal activity induced both *Scg2*-Luc and *Dio2*-Luc in astrocytes, which was inhibited by both co-expression of PKI and by ICER, a dominant negative member of the CREB superfamily (Fig. 6i,j). Collectively, these data implicate the cAMP–PKA–CREB pathway as an important mediator of activity-dependent gene expression in astrocytes.

**Neuronal activity boosts astrocytic metabolic capacity.** Given the number of AAR genes identified, the functional consequences are likely to be diverse. We focussed on one pathway: a large cluster of AAR genes centred on astrocytes' role in utilizing glucose to supply oxidizable substrates to neurons via the astrocyte–neuron lactate shuttle[34]. Remarkably, every sequential enzymatic and transport step on the pathway from glucose uptake, glycolysis, pyruvate-to-lactate conversion and lactate export included genes which were induced in astrocytes by synaptic activity (Fig. 7a; Supplementary Data 14). In contrast, components involved in mitochondrial import of pyruvate (pyruvate carriers) and NADH (the malate–aspartate shuttle), were not upregulated, suggesting that the reason for increased glycolytic capacity is to enable a boost in lactate export.

To investigate this, we used the pyruvate and lactate FRET probes Pyronic and Laconic[35]. By monitoring the rate of build-up of pyruvate or lactate upon acute blockade of lactate export using the monocarboxylic acid transporter (MCT) blocker AR-C155858 (ref. 35), we gained a measure of pyruvate production or lactate export rates. To ensure no acute effects of activity could confound the result, all experiments took place in TTX. We observed that in astrocytes co-cultured with neurons that had been previously active (BiC-treated for 16 h), the rate of pyruvate and lactate build-up upon MCT blockade was far faster than when the neurons were silent (TTX-treated; Fig. 7b–e). This indicates that neuronal activity has a long-lasting effect on the rate of astrocytic lactate export, consistent with the transcriptional changes observed.

Increased lactate export is likely to require increased glucose uptake and metabolism. To determine this we used the glucose FRET probe FLII[12]Pglu-700μδ6 (ref. 36), and used the method of measuring the rate of decline of the glucose FRET signal upon acute blockade of the glucose transporter with cytochalasin B[35]. We observed that prior activity resulted in an increased rate of decline of the glucose signal upon cytochalasin B treatment by ~two-fold, indicating that neuronal activity boosts glucose metabolism in astrocytes, consistent with elevated lactate export (Fig. 7f,g).

We next investigated the link between activity-dependent, CREB-mediated astrocytic gene expression (Fig. 6) and activity-induced increase in astrocytic metabolism. Sixteen of the 18 genes induced in the metabolic pathway highlighted in Fig. 7a have CRE full or half sites in the −5 to 1 kb region relative to

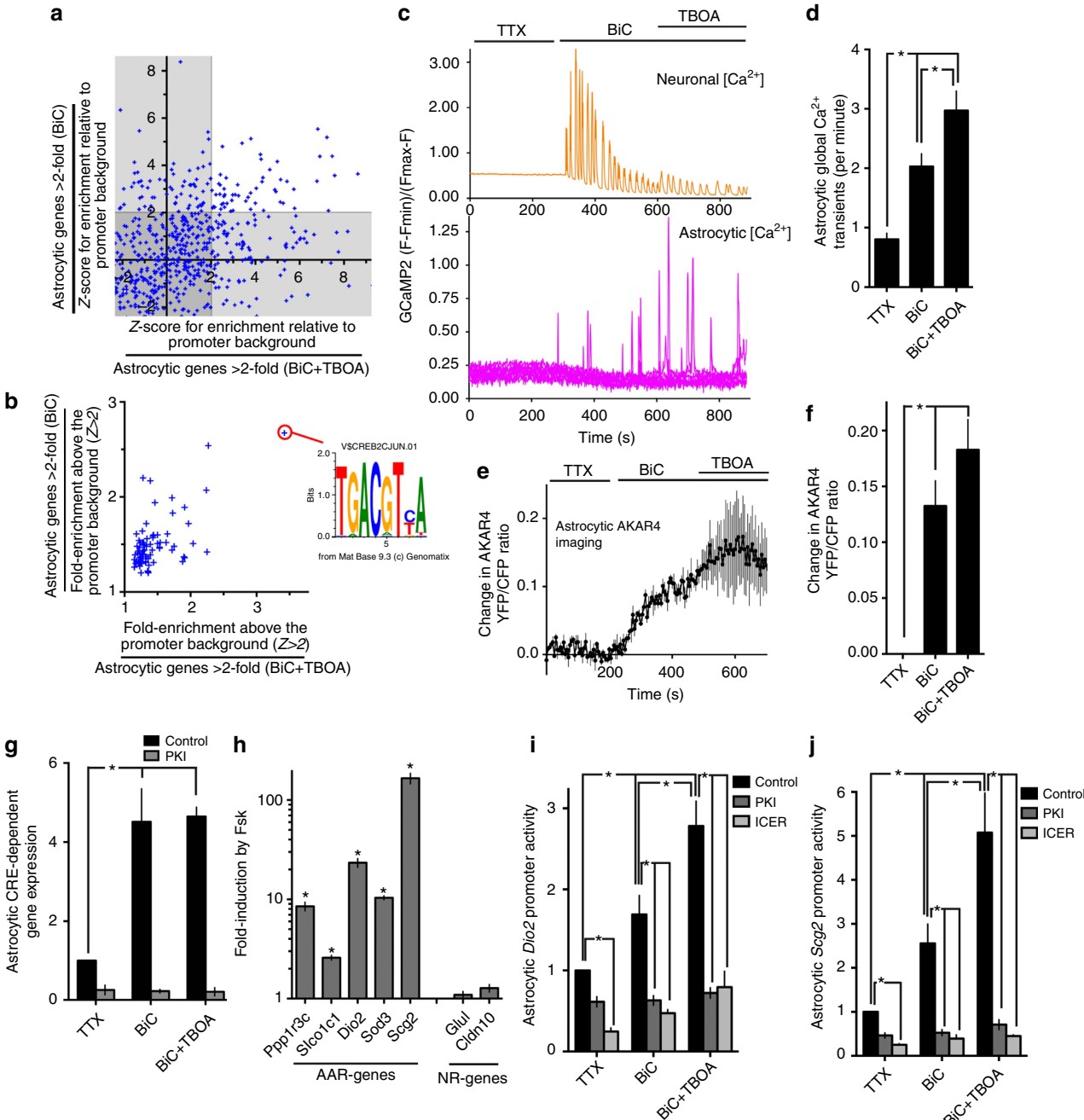

**Figure 6 | Astrocytic cAMP/PKA-dependent CREB activation contributes to activity-dependent astrocytic gene induction.** (**a,b**) Promoters of astrocytic activity-responsive genes (induced more than twofold) were analysed within Genomatix for enrichment in TRANSFAC Library matrices, with the $Z$-score for enrichment calculated. Matrices enriched ($Z > 2$) in the (overlapping) set of genes induced more than twofold by BiC and by BiC + TBOA are highlighted in **a**, and the degree of enrichment is shown in **b**, with the CREB family matrix highlighted (Supplementary Data 13). (**c,d**) Influence of neuronal activity induced by BiC ($\pm$ TBOA) on astrocytic $Ca^{2+}$ signals was measured. Left shows example traces of neuronal and astrocytic GCaMP2 imaging. Right shows influence of activity on astrocytic $Ca^{2+}$ transient frequency. *$P < 0.05$ (one-way analysis of variance (ANOVA) plus Tukey's *post hoc* test, $n = 5$, 32 cells total). (**e,f**) Influence of neuronal activity induced by BiC ($\pm$ TBOA) on astrocytic PKA activity (PKA FRET probe AKAR4 transfected into astrocytes before neuronal co-culture. (**e**) shows an example experiment showing the mean ($\pm$ s.e.m.) of several cells within a single field of view. (**f**) shows the quantitation. *$P < 0.05$ (one-way ANOVA plus Tukey's *post hoc* test, $n = 5$, 15 cells). (**g**) Neuronal activity-induced PKA-dependent CRE-mediated gene expression in astrocytes. Mouse astrocytes were transfected with a CRE-luciferase reporter (plus renilla control, + globin control or PKI vectors) before rat neurons were overlaid for 9 days. Neuronal activity was induced by BiC ($\pm$ TBOA) as described in Fig. 5a. Firefly luciferase activity was normalized to renilla transfection control. *$P < 0.05$ (two-way ANOVA plus Sidak's *post hoc* test, $n = 3$). (**h**) Forskolin treatment is sufficient to induce astrocytic activity-response (AAR) genes. Mixed rat neuron/mouse astrocytes were treated with forskolin (Fsk, 10 μM, 4 h) and gene expression measured by qPCR using mouse-specific primers. *$P < 0.05$ (two-way ANOVA plus Sidak's *post hoc* test, $n = 3$). (**i,j**) Activity-dependent induction of astrocytic *Dio2* and *Scg2* promoter activity is mediated by PKA and CREB. Mouse astrocytes were transfected with a *Dio2* (**i**) or *Scg2* (**j**) -luciferase reporter (plus renilla control, + control, PKI, or ICER vectors) before rat neurons were overlaid for 9 days. Neuronal activity was induced as described in Fig. 5a. Firefly luciferase activity was normalized to renilla control. *$P < 0.05$ (two-way ANOVA plus Sidak's *post hoc* test, $n = 3$). All error bars represent s.e.m.

the transcription start site (CREB Target Gene Database http://natural.salk.edu/CREB[37]). We investigated the effect of expressing the dominant negative CREB family member ICER on activity-dependent astrocytic glucose metabolism. While astrocytic ICER expression had no effect on glucose metabolism rates in astrocytes co-cultured with silent neurons, ICER strongly

inhibited the activity-dependent induction of astrocytic glucose metabolism (Fig. 7f,g). These data demonstrate that neuronal activity, via the activation of astrocytic CREB-mediated gene expression, boosts astrocytic glucose metabolic flux. Moreover, this is a pathway conserved in evolution: an identical experiment performed using human astrocytes also revealed a CREB-

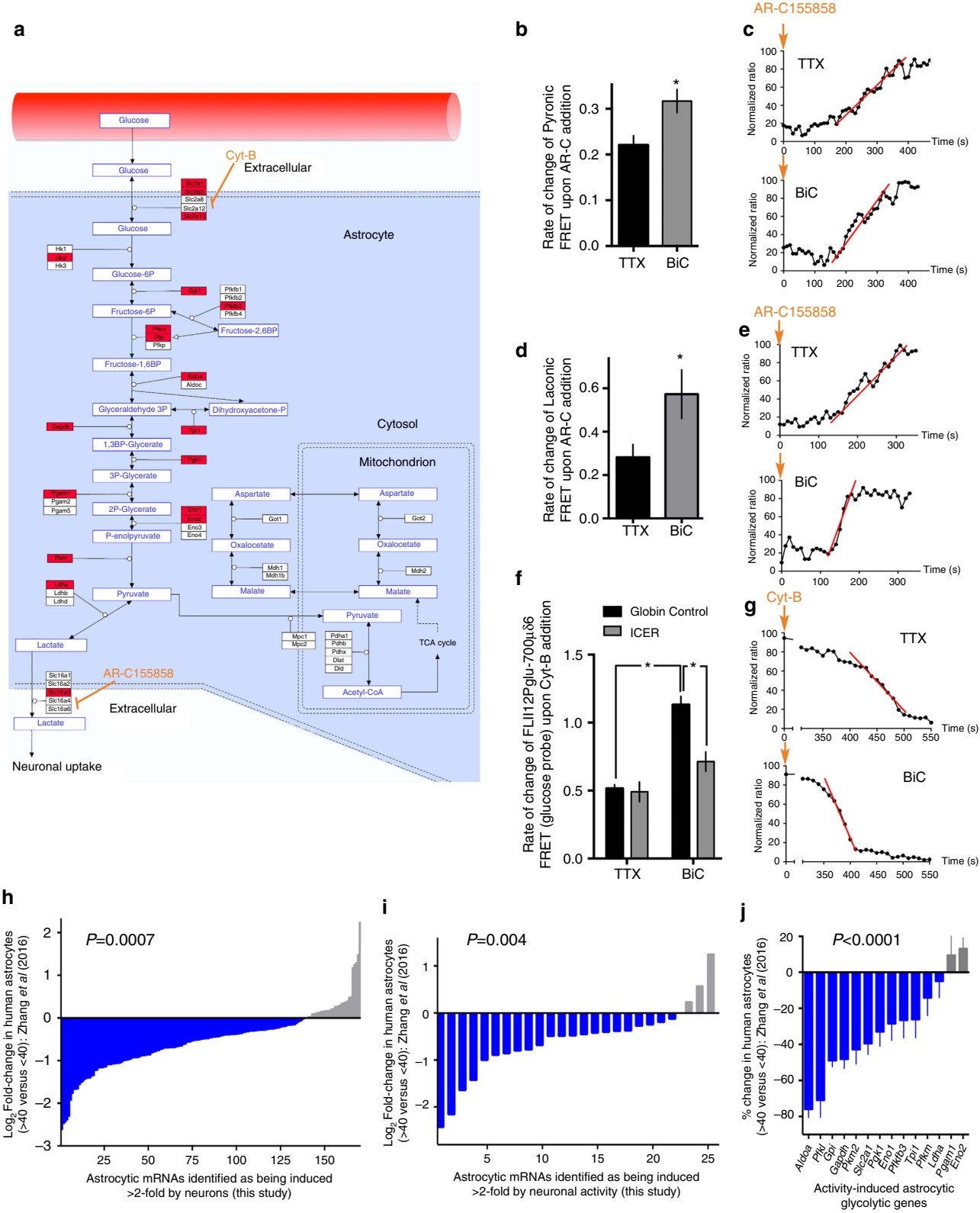

dependent induction of glucose metabolism in response to neuronal activity (Supplementary Fig. 7a). In addition, as well as CREB being necessary for activity-dependent increases in astrocytic glycolytic metabolism, it is also sufficient: astrocytes transfected with constitutively active CREB-VP16 exhibit elevated levels of glucose metabolism and lactate export (Supplementary Fig. 7b,c). Moreover, we observed that CREB-VP16 expression caused an increase in glycolysis (Supplementary Fig. 7d) that was not accompanied by changes in mitochondrial oxygen consumption rate (Supplementary Fig. 7e,f). This supports a model whereby the increase in astrocytic glucose metabolism serves to increase lactate export, rather than reflecting a strong increase in the astrocytes' own energy requirements.

Given the importance of gene regulation controlled by neurons and neuronal activity in astrocytic metabolic function, we mined an existing data set from Zhang et al.[18] to determine any age-related changes in NR or AAR genes in humans by comparing the 6 young samples (age 8–35 years) with the 6 older samples (age 47–63 years). We observed a significant decline in expression of both induced NR genes and AAR genes in astrocytes from older individuals (Fig. 7h; Supplementary Data 15; Fig. 7i; Supplementary Data 16). Within the group of activity-induced components of the astrocyte–neuron lactate shuttle there was also a significant reduction in astrocytes from the older cohort (Fig. 7j; Supplementary Data 19). Thus, astrocytic genes boosted by neurons and neuronal activity appear to decline with age, raising the possibility that their ability to support key homeostatic functions may also decline.

## Discussion

Studies into the signals and pathways that direct astrocytic fate specification from neural stem cells have revealed roles for CNTF or LIF-induced JAK/STAT signalling, BMP-SMAD signalling and Notch signalling[1]. In contrast, the mechanisms involved in astrocytic developmental maturation were not well understood[1]. Here we have not only uncovered the transcriptional changes that are associated with this transformation, but identified neuron-derived Notch signalling as a key mediator. This pathway is necessary and sufficient to promote one of the key functional roles of a mature astrocyte: glutamate uptake. Of note, expression of transporters involved in other neurotransmitters: GABA (Slc6a1 and Slc6a11), NAAG (Slc17a5) and biogenic amines (Slc29a2) are also induced, as are genes responsible for neurotransmitter metabolism: Glul and Glud1 (for glutamate), Abat (for GABA) and Maob (for biogenic amines). Thus, neurotransmitter uptake and metabolism capacity in astrocytes may be induced in a coordinated manner by neurons. Another

implication from this study is that neuron-derived signals may be required for the ongoing maintenance of astrocytic phenotype in vivo, raising the possibility that loss of neuron–astrocyte contacts in neurodegenerative disease has a knock-on effect on astrocytic phenotype, which could in turn amplify the pathological cascade.

Another finding of this study is the influence of neuronal activity on astrocytic metabolism. In astrocytes, energy-requiring processes of glutamate uptake and glutamine synthesis, increased in times of synaptic activity, trigger acute increases in astrocytic glucose uptake and utilization by the laws of mass action[38]. Much of this glucose utilization in astrocytes is non-oxidative: pyruvate is converted to lactate and exported by astrocytes where it can be taken up and used as a substrate in neurons, a pathway known as the astrocyte–neuron lactate shuttle (ANLS)[38]. Astrocytes are major contributors to brain glucose uptake, and represent the dominant route for uptake during periods of strong synaptic activity[39]. Moreover, astrocytic glucose uptake is of key importance to neurons, as evidenced by the severe neurological phenotype of mice haplo-insufficient for the astrocyte-specific glucose transporter Glut1 (refs 40–42). During execution of spatial memory tasks, extracellular levels of glucose decline, while lactate rises, in the hippocampus, with blockade of astrocytic glycogenolysis and neuronal lactate uptake leading to memory impairment, supportive of a key role for the ANLS in neuronal function[43], consistent with observations that astrocytic lactate export through Mct1/4 plays a key role in long-term memory[44]. Our study shows that synaptic activity triggers the transcriptional upregulation of astrocytic genes at every step of the ANLS, inducing glucose metabolism and lactate export. This may represent a homeostatic tuning of astrocytic metabolic flux to reflect the needs of nearby active neurons[45]. The observation that activity-dependent astrocytic ANLS genes decline with age in humans (Fig. 7j) raises the possibility that ageing astrocytes may be less equipped to metabolically support neurons, potentially leading to cognitive decline. Whether ageing astrocytes express activity-induced genes at lower levels (Fig. 7i) because they receive fewer neuronal signals, or are impaired in signal transduction properties, awaits investigation.

Moreover, future studies will likely uncover further functional changes to astrocytes in response to neuronal activity, since other notable gene clusters induced include those involved in thryroid hormone signalling (Slco1c1, Dio2 and Slc7a5)[46] and extracellular antioxidant capacity (Sod3 and Grx3). The latter may be important given the vulnerability of neurons to oxidative stress[47–49] and the importance of non-cell-autonomous astrocytic antioxidant support[13,50,51]. Of note, genes such as Dio2, Slco1c1 and Sod3 are very weakly expressed in neurons, and

**Figure 7 | Neuronal activity controls astrocytic metabolic flux via CREB activation.** (a) Schematic depicting astrocytic ANLS, mitochondrial NADH shuttling and pyruvate uptake genes (expressed >0.5 FPKM). Red indicates upregulated ($P_{adj}$<0.05) by BiC-induced synaptic activity. The schematic was created using WikiPathways[70]. (b–e) Prior neuronal activity boosts steady-state rate of astrocytic lactate export. Astrocytes were transfected with FRET probes for pyruvate (Pyronic, b,c) or lactate (Laconic, d,e) before rat neuronal co-culture for 9 days. The co-cultures were transferred into TTX for 22 h, before washout (+BiC) to induce activity for 16 h, while others were 'washed' in TTX-containing medium as a control. Before and throughout the imaging experiment, the co-cultures were re-treated with TTX to avoid any influence of ongoing neuronal activity. Pyronic and Laconic FRET ratios were measured before and after MCT blockade (AR-C155858, 1 μM). *P<0.05; n = 30 TTX, 23 BiC (b), n = 13, TTX, 12 BiC (d). Example traces are shown in c (Pyronic) and e (Laconic). (f,g) Astrocytes were transfected with the glucose FRET probe FLII12Pglu-700μδ6 plus either a ICER or control vector (β-globin) before rat neuronal co-culture for 9 days. Neuronal activity was induced as in b–e. Glucose probe FRET ratios were measured before and after glucose transporter blockade (cytochalasin-B, 20 μM) *P<0.05, two-way ANOVA plus Sidak's post hoc test (n = 33 TTX, 26 BiC (Con), n = 10, TTX, 13 BiC (ICER)). (h) Neuron-induced astrocytic genes (more than twofold) were cross-referenced to data in Zhang et al.[18], taking genes expressed >0.5 FPKM, and the fold change in expression in human astrocytes in the individuals <40 (n = 6) and >40 (n = 6) calculated. P = 0.0007 (age effect, paired t-test). (i) Activity-induced astrocytic genes (more than twofold) were cross-referenced to data in Zhang et al (2016), taking genes expressed >0.5 FPKM, and the fold-change in expression in human astrocytes in the individuals <40 (n = 6) and >40 (n = 6) calculated. P = 0.004 (age effect, paired t-test). (j) Activity-induced ANLS genes (a) were cross-referenced to data in Zhang et al.[18], taking genes expressed >0.5 FPKM, and the % change in expression in human astrocytes in the individuals <40 (n = 6) and >40 (n = 6) calculated. P<0.0001 (age-group effect, two-way ANOVA). All error bars represent s.e.m.

not altered by neuronal activity. Thus, this new type of 'synapse-to-nucleus' signalling likely has different effects on the neuro-glial unit than inter-neuronal signalling leading to gene expression changes.

## Methods

**Tissue cultures and stimulations.** Astrocytes and neurons were cultured from E17.5 CD1 mouse and E20.5 Sprague Dawley rat embryos as previously described[52,53]. Cortices or spinal cords were dissected, enzymatically digested with papain and mechanically dissociated using a 5 ml pipette. Mouse cortical and spinal cord astrocytes were obtained by growing cells at low density in DMEM containing 10% fetal bovine serum and were passaged twice, using Trypsin (both Life Technologies), before co-culturing them with neurons in a 24-well plate. These astrocytes are >99% GFAP positive and <0.1% NeuN positive[12]. For mixed–species co-cultures, rat neurons were plated on top of a confluent layer of DIV14 mouse astrocytes and both astrocyte monocultures and astrocyte–neuron co-cultures were subsequently kept in Neurobasal-A medium containing B27 (both Life Technologies), but devoid of serum. Note that the rat neuronal cells plated do contain a small number of non-neuronal cells (principally astrocytes), but since the focus of the study is the mouse astrocytic transcriptome, this was deemed acceptable. For Supplementary Fig. S5e, rat neuronal cells were plated in the continuous presence of the anti-mitotic AraC. Where used, pure neuronal cultures were generated as described[54] from E17.5 CD1 mouse embryos, with the anti-mitotic AraC added on the day of plating which restricts astrocyte numbers to <0.1%[13,54,55]. All cultures were used 8–10 days post neuron plate down. Cells were transferred into media containing 10% minimum essential media (MEM, Life Technologies) and 90% salt–glucose–glycine (SGG) medium containing 114 mM NaCl, 0.219% NaHCO$_3$, 5.292 mM KCl, 1 mM MgCl$_2$, 2 mM CaCl$_2$, 10 mM HEPES, 1 mM Glycine, 30 mM Glucose, 0.5 mM sodium pyruvate, 0.1% Phenol Red; osmolarity 325 mOsm/l either overnight or for 4 h before stimulations. To investigate the effect of synaptic activity on astrocytic gene expression, co-cultures were transferred into TTX-containing medium on DIV8 for 22 h to inhibit neuronal action potential firing. Subsequently, cells were washed with medium free of TTX and then kept in medium either containing Bicuculline (BiC, 50 µM, Tocris) to induce network bursting, or BiC and glutamate transporter inhibitor TBOA (DL-*three*-β-Benzyloxyaspartic acid, 50 µM, Tocris), to increase the half-time of glutamate uptake, for 16 h, after which RNA was collected. Control conditions were washed with TTX and left in medium containing either BiC or BiC/TBOA in the presence of TTX, or TTX alone. For experiments investigating the role of Notch signalling in astrocytic maturation, mouse astrocyte monocultures and mouse astrocyte/rat-neuron co-cultures were treated with the γ-secretase inhibitor DAPT (10 µM, Tocris) on DIV0 of neuron plate down. Fresh DAPT was added twice more and RNA collected on DIV9.

**Species-specific sorting of mixed species RNA-seq reads.** To generate RNA-seq data, barcoded RNA-seq libraries were prepared by Edinburgh Genomics using the Illumina TruSeq stranded mRNA-seq kit, according to the manufacturer's protocol (Illumina). The libraries were pooled and sequenced to 50 base paired-end on an Illumina HiSeq 2500 in high output mode (v4 chemistry). For single-species RNA-seq experiments sequencing was performed to a depth of ~50 million paired-end reads per sample, whereas for mixed-species RNA-seq a greater depth of ~150 million paired-end reads per sample was done.

Given a set of RNA-seq reads that may have derived from transcripts of both mouse and rat, we implemented a sorting procedure (called Sargasso: *Sargasso Assigns Reads to Genomes According to Species-Specific Origin*) to assign reads to their true species of origin. Essentially, Sargasso is a Python tool to disambiguate mixed-species RNA-seq reads according to their species of origin. Given a set of RNA-seq samples containing RNA-seq data originating from two different species, mapped, disambiguated reads are written to per-sample and -species-specific output BAM files. Sargasso is freely available (http://doi.org/10.5281/zenodo.206619) and is described briefly below.

In the balance between precision and recall, our initial strategy is conservative, in that it aims foremost to minimize the number of reads allocated to the incorrect species; but we simultaneously seek to maximize the number of reads that can be unambiguously assigned to the correct species. The sorting criteria detailed below were chosen to effect this goal. In this species-specific sorting (SSS) procedure, reads (or read pairs in the case of paired-end reads) are first mapped to the genomes of each species with version 2.4.0i of the STAR RNA-seq aligner[56]. At this stage multi-mapping alignments are allowed (–outFilterMultimapNmax 10000), but only those with an alignment score equal to the maximum (–outFilterMultimapScoreRange 0). Subsequently, for each RNA-seq read (or read pair) the alignments of that read to each genome are compared. If the read has alignments to the mouse genome, but no alignments to the rat genome exist, the read is provisionally assigned to the mouse; note, however, the further requirements given below for a read to be finally allocated to this genome. Similarly, if alignments to the rat genome exist, but there are no alignments to the mouse genome, the read is provisionally assigned to the rat. If alignments to both genomes exist, these alignments are examined in more detail. First, any read which aligns multiple times to either genome is discarded, since in this initial conservative

strategy their exclusion removes a potential source of ambiguity. Next, the number of mismatched bases between the mapped read and each genome is examined. If the number of mismatches is smaller for one species, the read is provisionally allocated to that genome. If the number of mismatches for each species is equal, a further check is made on the structure of the alignments; a successful alignment is required to span the full length of the read (without any clipping of bases) and, if the alignment spans an intron, at least 5 bases are required to align to the exons on either side of the boundary. If these criteria are satisfied for the alignment to one genome, but not to the other, then the read is provisionally assigned to the first genome. If the structural criteria are satisfied for the alignments to both genomes, the read is rejected as ambiguous; it cannot be assigned with confidence to one species rather than the other. The read is also rejected if the structural criteria are not satisfied for the alignments to either genome.

At this stage, a read has either been provisionally allocated to one species, or has been rejected. In the former case, a final set of checks are made on the provisional alignment. In this conservative strategy, these are that there should be no mismatches between the mapped read and the genome, and that the structural criteria outlined above are satisfied (if this has not already been confirmed). The outcome of this procedure is that all reads successfully assigned to one species or the other have a single, full-length alignment to that species' genome, with no mismatched bases. Subsequently, for each sample, per-gene read counts were summarized using featureCounts version 1.4.6-p2 (ref. 57). Relative expression levels of genes are expressed as fragments per million reads per kilobase of message (FPKM). Within the SSS workflow, only reads that are unambiguously attributed to a particular species are used as the denominator in the FPKM calculation. The value for the length of message for a particular gene refers to the maximum transcript length. Where gene length data are given, this refers to the number of nucleotides contained within the union of all exons of all transcripts of the gene, including 5′ and 3′ UTRs. Differential expression analysis on data sets was performed using DESeq2 (R package version 1.10.0)[58], with a significance threshold set at a Benjamini–Hochberg-adjusted P value <0.05, calculated within DESeq2).

For assessing the theoretical feasibility of SSS of mixed-species RNA-seq reads, we selected those genes in the mouse transcriptome, which were classified as protein coding in Ensembl version 82, but excluded predicted genes. Transcript sequences for all transcripts of these genes were created with the tool rsem-prepare-reference from the RSEM (RNA-Seq by Expectation-Maximization) software package[59]. For the analysis in Supplementary Fig. S1a, we created all possible theoretical 50 nucleotide paired-end reads, with insert size 150 nucleotide, obtainable from those transcripts (such read parameters were chosen as typical of the real RNA-seq data used in this study). This set of all possible 50 nucleotide paired-end read sequences were then mapped to the mouse and rat genomes using STAR, and the mapped reads assigned unambiguously to each species via the SSS procedure described above.

Before arriving at a final DGE data sets, we carried out an additional control by performing RNA-seq on a single-species rat co-culture of neurons and astrocytes, and determining whether the SSS workflow resulted in any rat reads being incorrectly called as mouse. This is important to rule out, since it could lead to erroneous DGE results. Performing this control revealed that for about 0.5% of rat genes >5% of their RNA-seq reads were incorrectly called as mouse, primarily due to imperfect annotation of the rat genome. We took a conservative approach and discarded any genes for which we estimated >10% of mouse reads within the mixed-species co-culture could be due to incorrectly called rat reads. This resulted in 268 out of the 16,629 genes expressed >0.5 FPKM in mouse astrocytes in the mixed-species co-culture being excluded from our analysis in Fig. 1c.

**Transfections and luciferase assays and plasmids.** Astrocytes and astrocyte–neuron co-cultures in a 24-well plate were transfected using Lipofectamine 2000 (Life Technologies) at 0.65 µg of DNA per well and 2.33 µl Lipofectamine 2000 per well, as described previously[60]. Cells were transferred into SGG(90%)/MEM(10%) containing insulin–transferrin–selenium (Life Technologies) and transfected for either 45 min (astrocyte monocultures) or 3 h (astrocyte–neuron co-cultures). To obtain astrocyte-specific transfection in co-cultures, the transfection of the astrocytes monoculture was performed 24 h before the addition of neurons to create the co-culture. This approach was taken when performing luciferase reporter assays, where Firefly luciferase-based reporter gene constructs (CBF1-Luc, CRE-Luc, Scg2-Luc and Dio2-Luc) were transfected along with a renilla expression vector (pTK-RL). Luciferase assays were performed using the Dual Glo assay kit (Promega) with Firefly luciferase-based reporter gene activity normalized to the renilla control (pTK-RL plasmid) in all cases. Plasmids encoding ICER[61], CBF1-VP16 (ref. 62), GCaMP2 (ref. 63), FLII$^{12}$Pglu-700µδ6 (ref. 36), Pyronic[64] and Laconic[65] have been described previously.

**Electrophysiological recordings.** Recording were performed as described[66,67]. Coverslips containing cortical neurons and astrocytes were transferred to a recording chamber perfused (at a flow rate of 3–5 ml min$^{-1}$) with an external recording solution composed of (in mM): 150 NaCl, 2.8 KCl, 10 HEPES, 2 CaCl$_2$, 1 MgCl$_2$, and 10 glucose, pH 7.3 (320–330 mOsm). Patch-pipettes were made from thick-walled borosilicate glass (Harvard Apparatus, Kent, UK), and when filled with the internal recording solution had tip resistances of 4–8 MΩ. A K-gluconate-

based internal solution was used for patching neurons composed of (in mM): K-gluconate 141, NaCl 2.5, HEPES 10, and EGTA 11; pH 7.3 with KOH. A KCl-based internal was used for patching astrocytes composed of (in mM): KCl 130, glucose 4, HEPES 10, EGTA 0.1, CaCl2 0.025, and sucrose 20; pH 7.2 with KOH. Astrocytes were voltage-clamped at $-80$ mV and any cells with a resting membrane potential $> -60$ mV upon break-in were discarded. To determine the maximal induced EAAT transport current, 200 µM L-Aspartate was bath applied followed by the addition of the high affinity EAAT inhibitor TFB-TBOA (20 µM) to ensure that any L-Aspartate-induced current was mediated by the transporter. AP5 (100 µM) was included in the external solution of all astrocyte recordings to block the activation of NMDARs by L-Aspartate. To determine the passive properties of astrocytes, cells were voltage-clamped from $-110$ to 0 mV in 10 mV steps and the steady-state current recorded. Neurons were voltage clamped at $-60$ mV and recordings were rejected if the holding current was $> -100$ pA or if the series resistance drifted by $>20\%$ of its initial value ($<30$ MΩ). No current was injected for passive current-clamp recordings and liquid-junction potential was not corrected for in any recording. Recordings were at room temperature (21 ± 2 °C) using a Multiclamp 200B amplifier (Molecular Devices, Union City, CA). Recordings were filtered at 5 kHz and digitized online at 20 kHz via a BNC-2090A/PCI-6251 DAQ board interface (National Instruments, Austin, TX, USA) and analysed using WinEDR 3.6 software (Dr John Dempster, University of Strathclyde, Glasgow, UK).

**Analysis of P301S tauopathy mouse model.** All procedures were performed in compliance with the UK Animals (Scientific Procedures) Act 1986 and institutional regulations, and approved by University of Edinburgh Local Ethical Review Board. The P301S mice were terminated at 5–6 months of age because of increasing hindlimb disability, in line with United Kingdom Home Office animal license regulations requiring humane killing. For genotyping, genomic DNA was isolated from ear tissue according to the Wizard SV Genomic DNA Purification System before quantitative PCR (qPCR) using Taqman techniques. Primers were hMAPT and mTERT (Life Technologies).

For fresh tissue for qPCR, C57BL/6 and P301S mice were sacrificed at 20 weeks of age. Sample size was estimated based on the variance of degeneration reported previously[22]. Mice were killed in a CO2 chamber before careful cervical dislocation. Cervical spinal cord (c5–c7) was removed and immediately frozen on dry ice and stored at $-80$ °C. To prepare tissue for IHC, mice were sacrificed at 20 weeks of age by a lethal intraperitoneal injection of 0.3 ml per 100 g body weight of sodium pentobarbital (euthetal). They were transcardially perfused with 30–50 ml of 1% PBS, then 50–100 ml 4% paraformaldehyde. Cervical spinal cord (c5–c7) was removed and post fixed in 4% paraformaldehyde overnight before cryoprotection in 25% sucrose solution. Spinal cords were frozen and cut on a cryostat (Leica) into 16 µm sections. Sections were mounted onto superfrost slides (VWR) and stored at $-80$ °C. All immunohistochemical analyses of spinal cord sections were performed blind.

**GCaMP2 and iGluSnFR imaging.** Astrocyte–neuron co-cultures were grown on glass coverslips (VWR) and imaged 9–10 days post neuron plate down. Cells were moved into an imaging chamber perfused with aCSF at 3–5 ml min$^{-1}$ and containing (in mM): NaCl (150), KCl (3), HEPES (10), glycine (0.1), CaCl2 (2), MgCl2 (1) and glucose (10) at pH 7.4. Imaging was performed at 37 °C on a Leica AF6000 LX using a DFC350 FX digital camera. To investigate the effect of synaptic activity on astrocytic glutamate exposure, astrocytes expressing iGluSnFR[68] and GCaMP2/mCherry-expressing neurons were transferred into TTX the night before imaging. On the day of imaging, cells were transferred into aCSF containing TTX. GCaMP2-positive neurons and iGluSnFR-positive astrocytes were brought into one field of view at ×20 and subsequently washed with TTX-free aCSF containing BiC to induce neuronal network bursting. To investigate the effect of glutamate uptake inhibition on astrocytic glutamate exposure, TBOA was applied to the cells. GCaMP2 and iGluSnFR were imaged using a standard GFP filter set at 1 Hz. To monitor astrocyte development, images of GFP$^+$ astrocytes in mono- and co-culture were taken every day between DIV0 and DIV10 after the addition of neurons using a standard GFP filter set at ×40. The cell outline was manually traced using Photoshop and the perimeter and area quantified using ImageJ. The ratio of perimeter:area was calculated to serve as a measure for morphological complexity.

**FRET probe imaging.** Mouse astrocytes were transfected with FRET probes for glucose (FLII12Pglu-700uDelta6), pyruvate (Pyronic) or lactate (Laconic) 24 h before co-culture with primary rat neurons. Cells were placed in tetrodotoxin (TTX) 100 nM at DIV9 following co-culture, and either kept in TTX, or washed with TTX-free medium containing bicuculline (50 µM) 24 h before imaging. All imaging was performed at 37 degrees Celsius in continuous perfusion with aCSF (composition: 150 mM NaCl, 3 mM KCl, 10 mM HEPES buffer, 0.1 mM glycine, 2 mM CaCl2, 1 mM MgCl2, and 10 mM glucose, pH 7.4). TTX (100 nM) was added to aCSF perfused in all conditions to exclude acute effects of altered neuronal activity on astrocyte metabolism at the time of imaging. Images were captured using a DFC350 FX digital camera as part of a Leica AF6000 LX imaging system. Images were acquired every 10 s. All probes were imaged with a standard

FRET CFP/YFP filter wheel, with excitation of CFP and measurements taken within the CFP and YFP emission spectra. For glucose measurement, the YFP/CFP ratio of FLII12Pglu-700uDelta6 FRET probe was used to determine intracellular glucose concentrations for individual astrocytes. Levels were measured at baseline in aCSF with TTX, before the addition of the glucose-uptake inhibitor cytochalasin B (20 µM) after 60 s, resulting in a reduction in intracellular glucose levels corresponding to the rate of glucose consumption. For pyruvate and lactate measurements, the CFP/YFP ratio of the Laconic and Pyronic FRET probes were used to determine intracellular lactate and pyruvate levels respectively. Levels were measured at baseline in aCSF, before the addition of the MCT inhibitor AR-C155858 (1 µM) resulting in an increase in concentration corresponding to lactate or pyruvate production. All FRET ratios were normalized by subtracting baseline and expressed as percentage of maximum. A linear least-squares fitting routine was used to determine the line of best fit and slopes calculated for the portion of the curve corresponding to the rate of consumption or production of substrate.

**Seahorse bioanalyser.** Cortical astrocytes from rats were seeded in XF 24-well cell culture microplates and infected with MOI 25 of Ad5-VP16-CREB, a serotype 5 adenovirus harbouring a constitutively active CREB or Ad5-Null as a control. After 18–24 h of viral expression, extracellular acidification rate (ECAR), proton production rate (PPR) and oxygen consumption rate (OCR) were measured using a Seahorse Bioscience XF-24 instrument (Seahorse Bioscience, North Billerica, MA). The instrument was calibrated following manufacturer's instructions. Cells were incubated with XF Assay Medium (Agilent) supplemented with 2 mM L-glutamine, 5.5 mM glucose and 1 mM sodium pyruvate for 30 min at 37 °C without CO2 to allow temperature and pH to reach equilibrium before the first measurement. Four measurements were done to measure each condition: baseline levels, 1 ug/ml of oligomycin (complex V inhibitor), 2 µM FCCP (mitocondrial uncoupler) and 0.4 µM rotenone (complex I inhibitor) plus 1 µM antimycin A (complex III inhibitor). All values were normalized to protein content of each well (measured by BCA assay) and expressed relative to basal level of control virus infected cells.

**RNA extraction RT-PCR and qPCR.** Total RNA extraction from astrocyte mono-cultures and astrocyte–neuron co-cultures was performed using the High Pure RNA Isolation Kit (Roche) and cDNA was subsequently created using the Transcriptor First Strand cDNA Synthesis Kit (Roche) using the following programme: 10 min at 25 °C, 30 min at 55 °C and 5 min at 85 °C. qPCRs were run on a Stratagene Mx3000P QPCR System (Agilent Technologies) using SYBR Green MasterRox (Roche) with 6 ng of cDNA per well of a 96-well plate, using the following programme: 10 min at 95 °C, 40 cycles of 30 s at 95 °C, 40 s at 60 °C and 30 s at 72 °C, with a subsequent cycle of 1 min at 95 °C and 30 s at 55 °C ramping up to 95 °C over 30 s (to measure the dissociation curve). Species-specific mouse primers were used (Table 1). Primers were designed to only pick up mouse transcripts in a mouse astrocyte/rat neuron co-culture, and were validated by running qPCRs with cDNA derived from either pure mouse or pure rat cultures, and discarded if they picked up any rat transcripts. H1f0 encodes a histone subunit and was used as loading control. Additional mouse primers were used without requiring species-specificity (Table 2). Human-specific primers were also used (Table 3).

**Immunohistochemistry.** For cell culture IHC, established protocols were employed[69]. Briefly, cells were fixed in 4% formaldehyde for 20 min at room temperature, washed with PBS and permeabilized with the detergent NP40 (Life Technologies). Cells were subsequently incubated in either rabbit anti-GFP (1:750, Life Technologies) or mouse anti-GFAP (1:400, Sigma) overnight at 4 °C. The next day, cells were washed with PBS and incubated with the appropriate secondary antibody at room temperature for 2 h. To stain neurons, cells were incubated with the pan neuronal marker Neuro-Chrom, directly conjugated to Cy3 (Merck Millipore). Cells were then mounted using the mounting medium Vectashield (Vector Labs).

For spinal cord IHC, fixed spinal sections from 20-week-old C57BL/6 ($n = 5$) and P301S ($n = 5$) mice were stained for AT8, GFAP, Aldh1l1 and ChAT. Spinal cord slides were allowed to defrost and air dry before two 5 minute washes in 1% PBS (137 mM NaCl; 2.7 mM KCl;10 mM Na2HPO4; 1.8 mM KH2PO4; pH = 7.4 adjusted with HCL), on a shaker. Sections were blocked and permeabilized with 3% normal goat serum or 3% normal horse serum (S-1000 or S-2000, Vector Laboratories) and 0.2% Triton-X (X-100, Sigma) in PBS (~300 µl per slide) for 1 h. They were incubated with primary antibodies with 1% blocking serum in 0.2% Tx-PBS over-night. Three 10 min washes in 1% PBS on a shaker were followed by 2 h incubation with secondary antibodies, as required, with 1% NGS, Bis-benzamide (1:4000, B1155, Sigma Aldrich) in 1% PBS. Slides were washed in 1% PBS (10 min) and TNS (Tris non-saline solution, 50 mM Tris, pH 7.4 with Nitric acid) twice for 15 min, before mounting in fluorosave reagent (345789, Millipore). Primary antibodies used were mouse monoclonal anti-phospho-tau (AT8, 1:1000, Autogen Bioclear); rabbit polyclonal anti-GFAP (1:1000, Sigma C9205); polyclonal goat anti-ChAT (1:200, Millipore AB144P), rabbit polyclonal anti-Aldh1l1 (1:1,000, Abcam, ab190298). Secondary antibodies were donkey anti-mouse Alexa 555 (1:1,000 Invitrogen); Donkey anti-goat Alexa

**Table 1 | Species-specific mouse qPCR primers.**

| Gene | Fwd | Rev |
|---|---|---|
| H1f0 | 5′-GTTTGTCTTCCAAGACTTTCTT-3′ | 5′-CTTTGCCCCTTTAGACAATGGG-3′ |
| Dio2 | 5′-CCCTTCTGAGCGAATTGATCCA-3′ | 5′-CACATCGTAAGTATGTATCTGGG-3′ |
| Slc1a2 | 5′-TATCATCTCCAGTTTAATCAC-3′ | 5′-TTCATTCAACATGGAGATGACC-3′ |
| Hes5 | 5′-TGCAGAGTTGTCATTTGGGG-3′ | 5′-AACGGGCCCTGAAGAAAGT-3′ |
| Hey2 | 5′-GAATGTAACGTAGCACAAGATCAG-3′ | 5′-AGGTCTTTCGACTTAATTTCCC-3′ |
| Glul | 5′-GAGATTGACATTTCCACTGTTGG-3′ | 5′-ATCCATCAGGTGACGCGGTGAG-3′ |
| Cldn10 | 5′-CCCACACTTCAAGCCATGAGAT-3′ | 5′-GGAAGGAGCCCAGAGCGTT-3′ |
| Rlbp1 | 5′-CGTGGAAGGCAGAGTTAAAGGC-3′ | 5′-CAAGGATCACATCCAAGATGGG-3′ |
| Adora2b | 5′-ACTGGCCGATCCTCACTGTGAA-3′ | 5′-GAATCAATTCAAGCTGCCACCA-3′ |
| Slco1c1 | 5′-GATCCAGACCCTTGCGAACAT-3′ | 5′-GATATCCGACTGTAAAGGATGG-3′ |
| Ppp1r3c | 5′-CAGATGTGGACTGTGTCTACA-3′ | 5′-ATCCTCCCATTAGCGTGATAA-3′ |
| Slc6a11 | 5′-CTATGATGCCCCTCTCTCCAC-3′ | 5′-CTGTCACAAGACTCTCCACG-3′ |
| Grm3 | 5′-CATGTTGTTTGCCATTGATGAA-3′ | 5′-ATGCTCTGACAAACTCCAGTGAC-3′ |
| Fmo1 | 5′-TCGTCTTTGCGACTGGATATACT-3′ | 5′-GCTTGATGAGGCCAATCACA-3′ |
| Cyp4f15 | 5′-CACACAGTGACTCCCTGCAC-3′ | 5′-TCTGGGCCAAAGGATGCT-3′ |
| 2900052N01Rik | 5′-ACAATCCAAATCTACCCACGAG-3′ | 5′-TGGGATTGTAGATTGTGCTGTC-3′ |
| Slc1a3 | 5′-CAAGACACTGACACGCAAGGAC-3′ | 5′-CTTAACATCTTCCTTGGTGAGGC-3′ |
| Rorb | 5′-AGCATAGATTCCGGTCAGC-3′ | 5′-GAGTTCTTCCATGGTGTACTGAC-3′ |
| Ppp1r3g | 5′-GATGCCTGATCCTCTCTTG-3′ | 5′-ACTGATCACTCGGCCAG-3′ |
| Tmem100 | 5′-GCCCACACTGCTCTAACTC-3′ | 5′-CACTCCCTAAACGTTTAACAGG-3′ |
| Alp/ | 5′-GGCAATGAGGTCACATCC-3′ | 5′-CTGGTGGCATCTCGTTATC-3′ |

Fwd, forward; qPCR, quantitative PCR; Rev, reverse.

**Table 2 | Non species-specific mouse qPCR primers.**

| Gene | Fwd | Rev |
|---|---|---|
| Aldh1l1 | 5′-CATCCAGACCTTCCGATACTTC | 5′-ACAATACCACAGACCCCAAC-3′ |
| Fmo1 | 5′-CATCTGCCAAAACCAACTCTG | 5′-TGGCGGTGGTAATGTAGTTG-3′ |
| Grm3 | 5′-CCAAGCTCTGTGATGCAATG | 5′-CCGTCTCCGTAAGTGTCAAAC-3′ |
| 2900052N01Rik | 5′-ACAATCCAAATCTACCCACGAG | 5′-TGGGATTGTAGATTGTGCTGTC-3′ |
| Gfap | 5′-GCAAAAGCACCAAAGAAGGGGA | 5′-ACATGGTTCAGTCCCTTAGAGG-3′ |
| Cyp4f15 | 5′-CCCCAGTAAGCATGAGGATG | 5′-CAAACATGAAGGTGTCAGCC-3′ |
| Slc6a11 | 5′-CTATGATGCCCCTCTCTCCAC | 5′-CTGTCACAAGACTCTCCACG-3′ |

Fwd, forward; qPCR, quantitative PCR; Rev, reverse.

**Table 3 | Species-specific human qPCR primers.**

| Gene | Fwd | Rev |
|---|---|---|
| SLC1A2 | 5′-TTATTTATGTTCGGTTTGCCT-3′ | 5′-CTAGGACGATGAGATGATGACT-3′ |
| SLC1A3 | 5′-ACCTGCCCTCTGTTTCC-3′ | 5′-ATGAATAATCCCACTCCTGG-3′ |
| AQP4 | 5′-GAGAGTCGTCACACCAGTG-3′ | 5′-TCCCAGCCAGGAAGTAACTA-3′ |
| RPL13A | 5′-CCACTACCGGAAGAAGAAACAG-3′ | 5′-CAGGGCAACAATGGAGG-3′ |
| HES5 | 5′-TCTTCTGCCAAGTGTCTGAC-3′ | 5′-CCGGCACTACAAATATCATAGA-3′ |
| HEY2 | 5′-TGAGAGAGTCGTGTTTCGTAAG-3′ | 5′-CAACTTGAAAATTATTTTCAGCAG-3′ |

Fwd, forward; qPCR, quantitative PCR; Rev, reverse.

Fluor 488 (1:1000, A11055 Life Technologies), Goat anti-mouse Alexa Fluor 568 (1:1,000, A11004 Life Technologies) and donkey anti-rabbit Alexa 488 (1:500, Jackson ImmunoResearch).

For image analysis, Tau (AT8) was imaged using a Zeiss Axiovision microscope, ( × 20 objective with exposure time 800 ms) and Axiovision 4.8 software via a digital camera and stitched together using AxioVision software panorama tool. For imaging of C57BL/6 spinal cord exposure was increased to enable stitching. The imaging of GFAP and ChAT was carried out on a Zeiss LSM710 laser scanning confocal microscope. Images were taken at × 20 magnification. Quantitative analysis for motor neurons using ChAT stain was performed on the ventral horn of six spinal sections from each animal. The analysis was carried out at × 63 magnification under 152 μm × 152 μm grid. Independent sample t-test was performed for statistical significance at α = 0.05. Aldh1l1 images were taken with a DFC350 FX digital camera on a Leica AF6000 LX microscope at × 20 magnification.

**Statistical analysis.** Statistical testing of the RNA-seq data is described in that section. Other testing involved a two-tailed paired Student's t-test, or a one- or two-way analysis of variance followed by Sidak's post hoc test, as indicated in the legends. For t-tests, variance was generally found to be similar, abrogating the need for Welsh's Correction. Sample sizes were calculated using standard power calculations, requiring an effect size of 30% at 80% power. Throughout the manuscript, independent biological replicates are defined as independently performed experiments on material derived from different animals.

**Data availability.** The RNA-seq read sorting procedure (Sargasso: *Sargasso Assigns Reads to Genomes According to Species-Specific Origin*) is a Python tool to disambiguate mixed-species RNA-seq reads according to their species of origin. Given a set of RNA-seq samples containing RNA-seq data originating from two

different species, mapped, disambiguated reads are written to per-sample and -species specific output BAM files. Sargasso is freely available (http://doi.org/10.5281/zenodo.206619). All the RNA-seq data that support the findings of this study have been deposited in the European Bioinformatics Institute depository (accession code: E-MTAB-5514). All the other data are available from the corresponding author upon reasonable request.

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

## Acknowledgements

We acknowledge the wealth of informative gene expression data published by the laboratory of Ben Barres, which formed part of our meta-analyses in this study. We thank Paul Skehel and David Price for comments on the manuscript. We are very grateful to Paulo Sassone-Corsi, Richard Maurer, Ronald Evans, Wolf Frommer and L Felipe Barros for plasmids. RNA-seq raw reads were generated by Edinburgh Genomics, The University of Edinburgh. Edinburgh Genomics is partly supported through core grants from NERC (R8/H10/56), MRC (MR/K001744/1) and BBSRC (BB/J004243/1). This work is also supported by the Medical Research Council, the Wellcome Trust, the Biotechnology and Biological Research Council, the NPlast European Commission Initial Training Network, and Biogen through a Edinburgh University-Biogen Collaborative Research Initiative.

## Author contributions

G.E.H. conceived the project and wrote the manuscript. D.J.A.W., S.C., A.Z., R.M., E.G. and G.E.H. directed the research. P.H., N.M.M., Z.J., A.T., P.B., J.M., D.H., S.M., S.S.T., A.E.-P., D.H., M.T. performed the wet lab experiments. O.D., S.H. and T.I.S. developed and tested the SSS workflow.

## Additional information

**Competing interests:** The authors declare no competing financial interests.

DOI: 10.1038/ncomms16176    OPEN

# Author Correction: Neurons and neuronal activity control gene expression in astrocytes to regulate their development and metabolism

Philip Hasel, Owen Dando, Zoeb Jiwaji, Paul Baxter, Alison C. Todd, Samuel Heron, Nóra M. Márkus, Jamie McQueen, David W. Hampton, Megan Torvell, Sachin S. Tiwari, Sean McKay, Abel Eraso-Pichot, Antonio Zorzano, Roser Masgrau, Elena Galea, Siddharthan Chandran, David J.A. Wyllie, T. Ian Simpson & Giles E. Hardingham

Nature Communications 8:15132 doi:10.1038/ncomms15132 (2017); Published 2 May 2017; Updated 6 Feb 2018

Michel Goedert, who developed the Thy1-P301S transgenic mouse, was inadvertently omitted from the Acknowledgments section of this Article. The Acknowledgements should have included the following:

'We thank Michel Goedert for providing the Thy1-P301S transgenic mouse that was used in this study.'

