## [Peer Review File · Nature Communications]

Reviewers' comments:

Reviewer #1 (Remarks to the Author):

In the manuscript entitled "Neurons and neuronal activity control gene expression in astrocytes to regulate their development and metabolism," Hasel and colleagues utilize a species specific co-culture paradigm to probe how neurons and neural activity regulate astrocyte gene transcription. By co-culturing mouse astrocytes and rat neurons, performing RNA-seq from total isolated mRNA and sorting the reads based on species assignment, the authors demonstrate the ability to identify mouse astrocyte genes up and down regulated due to the presence of neurons. This platform was utilized in conjunction with activity blockers and pharmacology to tease apart neuron-induced and activity-induced astrocyte gene expression changes. In mining this data set, the authors produce and test hypotheses that astrocyte maturation is controlled by cAMP/PKA pathways and that synaptic activity leads to transcriptional and functional changes to glucose metabolism and lactate export. Furthermore, the authors produce evidence that astrocyte gene expression is dynamic depending upon the status of neuron health and relates to aging and neurodegenerative diseases.

The study produces a wealth of RNAseq data of high interest for the neuroscience field and in particular to neurobiologists who want to understand the astrocyte-neuron interactions. The majority of the experiments in the study are in vitro; however, this is not necessarily a weakness because answering the questions posed is technically challenging (near impossible) in vivo. Hence, the study tackles very relevant questions and is in line with the objectives of Nature Communications and in general I am in favor of its publications. There are, however, a few minor points that need to be addressed prior to publication. These concerns are detailed below.

- 1) In figure 3E, the authors mention that they are normalizing to Aldh1L1. Does Aldh1L1 levels change at all between WT and P301S, and if so this may skew the results. Please show Aldh1L1 levels.
- 2) In figure 3A, the authors show that neuron-dependent astrocyte gene expression can be repressed following removal from a P7 brain. P7 is the beginning of synapse formation and these synapses are fairly immature. Does astrocyte gene expression return to a baseline from mature astrocytes (e.g. P50 astrocytes)?
- 3) In figure 4G, the authors demonstrate that a constitutively active form of Notch signaling by expression of CBF1 can increase glutamate transporter currents in the absence of neurons. Can CBF1 overexpression drive the increase of all Notch signaling transcripts in the absence of neurons? This may answer the question posed on page 8, which asks which genes are direct or indirect notch responsive genes.
- 4) In figure 5G, the authors ask which astrocytic genes are regulated by NMDA synaptic activity by blocking NMDARs with MK-801. However, the authors show in Supplemental Table 12, that Grin2C (NR2C) is highly induced in astrocytes by neuron contract. Therefore, blockade of NMDARs by MK-801 may have astrocyte cell-autonomous affects and compound the results in Figure 5G. Please try blocking synaptic activity via an alternative method.
- 5) The authors do mention that a pitfall to figure 6H, is that treatment of the PKA inhibitor PKI, will affect neurons and astrocytes. They state "it was not possible to pharmacologically inhibit PKA to assess its role in activity dependent astrocytic gene expression because the inhibitors also inhibited neuronal activity itself." The authors should explore using a dominant negative of PKA expressed only in astrocytes to address this limitation.
- 6) The authors use forskolin to drive adenylyl cyclase and measure AAR genes in astrocytes when cultured with neurons. Can forskolin alone regulate AAR gene expression in astrocyte only cultures?

Reviewer #2 (Remarks to the Author):

In this manuscript the authors co-cultured mouse cortical astrocytes with rat cortical neurons and performed RNASeq on the cultured cells. They separated mouse from rat transcripts and identified several neuron-induced genes in astrocytes. They also co-cultured rat neurons with human astrocytes and found that transcriptional responses in human cells mirror those of mouse astrocytes. Freshly explanted astrocytes lose neuron induced genes after 4 days in culture and addition of neurons restores expression of these genes. Using a P301S transgenic mouse that expresses mutated Tau and develops Tauopathy, the authors found that neurodegeneration triggered by this mutated neuron-specific transgene leads to a decline in the astrocytic expression of neuronally-regulated genes in vivo. Hundreds of astrocytic genes were found to be acutely regulated by glutamatergic synaptic activity via mechanisms involving cAMP/PKA-dependent CREB activation. The groups of astrocytic genes induced by neurons or neuronal activity both show age-dependent decline in humans.

Overall, the concept that neurons regulate gene expression in astrocytes is important and has significant implications for various physiological and pathological conditions. The manuscript is well written and contains a lot of data. I have some concerns that affect the impact of the manuscript as follows:

1-First, the rationale for using mixed species is not really convincing. Why not instead use transgenic animals that express a fluorescent protein in astrocytes and a different fluorescent protein in neurons and separate these cells to very high purity by flow?. Indeed, the use of mixed species carries several hazards by itself. A major problem is that rat cortical neurons could cause changes in mouse astrocytes different than those induced by mouse neurons because of antigenic differences and the likely presence of small numbers of contaminating immune cells such as microglia and blood cells. In this regard the system used in this manuscript does not account for such potential problems.

2-Most of the data generated in this manuscript is based on cells in culture. It is well established that cells in culture change their gene expression profiles and are different from cells in vivo. This puts into question the relevance of the data shown in the manuscript. Even experiments showing the effects of neuronal activation, can now be done in vivo (by optogenetics for example).

3-The experiment described on page 5 To assess the usefulness of the mixed species approach in avoiding cell-type contamination associated with physical separation approaches, by simulating a sorting process that achieved 95% purity, and 'contaminating' the astrocytic mouse reads with the sorted rat (primarily neuronal) reads such that they formed 5% of total reads is not appropriately done. Instead, the authors should actually mix the actual cells in culture to contain an 95% to 5% ratio then perform the RNASeq and see if that affects the results.

4-Cortical and spinal cord astrocytes are different. Therefore it is not appropriate to use spinal cord cells in the experiments described in figure 3. Cortical cells should be used instead.

Reviewer #3 (Remarks to the Author):

This study presents interesting and original data about the influence of neurons and neuronal activity on the regulation of gene expression in astrocytes using RNA-Seq. The authors overcome the issues of cell-type contamination and artifacts due cellular stress during physical separation of cells by establishing an astute in silico sorting procedure with co-cultures of neurons and astrocytes from different species. In addition, convincing control experiments were carried out to validate their mixed species approach during the different experimental tests (mouse and human astrocytes, and the influence of synaptic activity).

This is an extensive piece of work and a well written manuscript which conveys important messages about neuron-to-astrocyte communication and the role of neuronal activity in shaping the functional properties of astrocytes such as neurotransmitter uptake and metabolic homeostasis. In addition,

specific signaling mechanism were identified in each case and validated with physiological approaches .

This paper should be of great interest to the reader

The presentation of the results, however, is too narrative and could benefit from more quantitative descriptions. It would be instructive to have access to experimental parameters such as read counts, percentages, cutoffs, etc, and to how genes were identified and regulated in each experiment without having to check each table. Perhaps a summary table could be embedded in the results section displaying these information.

The statistical analyses (or their descriptions) also need to be extended. For example, it isn't clear whether data from the RNA-seq analysis are coming from one sample or whether they have been repeated using replicates from the same or independent cultures. Likewise, it is unclear how the statistical significance is established when comparing gene sets for NR- or AAR-genes to gene sets published in previous paper. Proper statistical test should be used and described.

Minor comments:

In the assessment of the species-specific sorting RNA-Seq workflow, the authors evaluate the attribution of reads to species-specific genomes using only mouse RNA-seq data sets. What would have been the outcome of these experiments with data sets from the rat instead ? Along this line, would the findings described in the subsequent experiments be the same if the experiments were conducted with mouse instead of rat neurons plated on top of rat instead of mouse astrocytes ?

The expression threshold (in FPKM) to exclude genes from analyses are not the same across all experiments. Why is that ? How would it influence the outcome of each analysis ?

For astrocytic genes induced by synaptic activity the cutoff was >1.3 whereas it was >2-fold for most of the other experiments of the paper. Could the author explain the relevance of this change ?

Samples label "young" (age 8-35) compared to "older" (47-63) in figure 7 are in conflict with the terminology "mature" (age 8-63) compared to "fetal" (18-19 gestational weeks). This is confusing. I suggest that the authors choose more harmonious descriptive terms.

There is already a lot of material with the various comparisons done in this manuscript. In the discussion, pointing out all the advantage of mixed species RNA-seq is pertinent but I don't see the point of commenting experiments which are not described in this paper. This should be removed.

Reviewers' comments:

Reviewer #1 (Remarks to the Author):

In the manuscript entitled “Neurons and neuronal activity control gene expression in astrocytes to regulate their development and metabolism,” Hasel and colleagues utilize a species specific co-culture paradigm to probe how neurons and neural activity regulate astrocyte gene transcription. By co-culturing mouse astrocytes and rat neurons, performing RNA-seq from total isolated mRNA and sorting the reads based on species assignment, the authors demonstrate the ability to identify mouse astrocyte genes up and down regulated due to the presence of neurons. This platform was utilized in conjunction with activity blockers and pharmacology to tease apart neuron-induced and activity-induced astrocyte gene expression changes. In mining this data set, the authors produce and test hypotheses that astrocyte maturation is controlled by cAMP/PKA pathways and that synaptic activity leads to transcriptional and functional changes to glucose metabolism and lactate export. Furthermore, the authors produce evidence that astrocyte gene expression is dynamic depending upon the status of neuron health and relates to aging and neurodegenerative diseases.

The study produces a wealth of RNAseq data of high interest for the neuroscience field and in particular to neurobiologists who want to understand the astrocyte-neuron interactions. The majority of the experiments in the study are in vitro; however, this is not necessarily a weakness because answering the questions posed is technically challenging (near impossible) in vivo. Hence, the study tackles very relevant questions and is in line with the objectives of Nature Communications and in general I am in favor of its publications. There are, however, a few minor points that need to be addressed prior to publication. These concerns are detailed below.

We were very pleased that the reviewer felt our study tackles very relevant questions, and contains data of high interest for the neuroscience field. The points are addressed below.

1) In figure 3E, the authors mention that they are normalizing to Aldh1L1. Does Aldh1L1 levels change at all between WT and P301S, and if so this may skew the results. Please show Aldh1L1 levels.

We chose to normalize the genes to Aldh111 because it is a specific marker for astrocytes in multiple species^{1, 2} whose expression is not influenced by neuronal contact (Fig. 1c), nor is it altered in vivo following inflammatory (LPS) or ischemic (MCAO) injury³. We therefore predicted that these other neuronally-regulated astrocyte-specific genes would be reduced *relative* to Aldh111. We now make this clearer. As to whether Aldh111 levels rise in astrocytes in the P301S mouse, we have now studied this and have no evidence that this is the case. We looked at the relative levels of Aldh111 in WT vs. P301S samples based on the cycle threshold of amplification of identical quantities of starting RNA, and found no difference (new Fig. S4e). We also performed immunohistochemical analysis of Aldh111 protein levels in spinal cord sections and again found no evidence for a difference when comparing WT vs. P301S mice (new Fig. S4f,g) .

2) In figure 3A, the authors show that neuron-dependent astrocyte gene expression can be repressed following removal from a P7 brain. P7 is the beginning of synapse formation and these synapses are fairly immature. Does astrocyte gene expression return to a baseline from mature astrocytes (e.g. P50 astrocytes)?

Dissociating mature brains and sorting cells such that they can then adhere to tissue culture plastic and retain viability for ongoing tissue culture is very challenging. While sorting the cells themselves was possible, consequent survival rates were poor in vitro. Thus, we were unable to determine the reversibility of neuron-induced astrocytic gene expression at this stage. However, the P301S mouse experiment suggests that in vivo, loss of neurons does have this effect in adult mice.

3) In figure 4G, the authors demonstrate that a constitutively active form of Notch signaling by expression of CBF1 can increase glutamate transporter currents in the absence of neurons. Can

CBF1 overexpression drive the increase of all Notch signaling transcripts in the absence of neurons? This may answer the question posed on page 8, which asks which genes are direct or indirect notch responsive genes.

We performed over-expression experiments using a strong activator of Notch signaling (intracellular domain of Notch1: Notch1-IC) to compensate for fairly low transfection efficiencies obtained in astrocytes. Gene expression was analysed in Notch1-IC transfected astrocytes at 48h, a timepoint chosen since it should be adequate time to observe direct effects, but potentially not downstream consequences. We observed that, while *Hes5* and *Hey2* were strongly induced, the other genes are not, suggesting that they are indeed downstream of Notch signaling. We now state this and show the data in Fig. S5b.

4) In figure 5G, the authors ask which astrocytic genes are regulated by NMDA synaptic activity by blocking NMDARs with MK-801. However, the authors show in Supplemental Table 12, that Grin2C (NR2C) is highly induced in astrocytes by neuron contract. Therefore, blockade of NMDARs by MK-801 may have astrocyte cell-autonomous affects and compound the results in Figure 5G. Please try blocking synaptic activity via an alternative method.

We apologize for not making that Table clearer. The column "Astro/neuro-co:neuro-mono" refers to expression levels in single species mouse astrocyte/neuron co cultures, compared to mouse neuronal mono-cultures. In other words, it is genes expressed more strongly in astrocytes than neurons. It does not refer to any influence of neuronal contact on astrocytic expression. Thus, this high number is simply because cortical neurons express only tiny amounts of *Grin2c* (<0.05 FPKM).

Nevertheless, the broader question as to whether astrocytic NMDARs could be mediating activity dependent gene expression is one we are happy to address. From our RNA-seq data we observe that neurons express around 50 times higher levels of obligate NMDAR subunit gene *Grin1* than cortical astrocytes in co-culture (50 FPKM vs. 1 FPKM). Regarding the glutamate-binding GluN2 subunits (*Grin2* genes), their combined expression levels in cortical neurons is approximately 25 times higher than that in astrocytes. Thus, the capacity for astrocytic NMDARs to mediate meaningful responses to glutamate exposure, leading to transcriptional changes, may be very limited. To test this directly, we treated mixed species co-cultured astrocytes with exogenous glutamate at a concentration (20 μ M) >5-fold higher than the EC-50 of the lowest affinity GluN2, all in the presence of TTX to prevent indirect influences due to neuronal firing. We found that glutamate (in the presence or absence of NMDAR antagonist MK-801) failed to influence the expression of astrocytic activity-responsive genes (Fig. S6b). In contrast, adenosine receptor agonist ATP induced a subset of these genes (Fig. S6b). Thus, the NMDAR, and glutamate itself, is unlikely to play a major role in synapse-to-astrocyte signaling in the control of gene expression, strongly suggesting that MK-801 is unlikely to be having astrocyte cell-autonomous affects.

As an aside, these investigations suggest that the influence of TBOA on activity-dependent astrocytic gene expression is likely to be due to its effect in prolonging action potential bursts in the neurons (Fig. S6a), rather than in prolonging astrocytic glutamate exposure. We now make this clear.

5) The authors do mention that a pitfall to figure 6H, is that treatment of the PKA inhibitor PKI, will affect neurons and astrocytes. They state "it was not possible to pharmacologically inhibit PKA to assess its role in activity dependent astrocytic gene expression because the inhibitors also inhibited neuronal activity itself." The authors should explore using a dominant negative of PKA expressed only in astrocytes to address this limitation.

This is a misunderstanding. PKI is not a drug but a protein (gene name *Pkia*: protein kinase (cAMP-dependent, catalytic) inhibitor alpha) which directly inhibits the catalytic subunit of PKA. This is indeed transfected only into astrocytes for these experiments. No pharmacological inhibitors of PKA are used in these experiments, for the reasons that the reviewer cites. We now make this clearer in the text to emphasize what PKI actually is.

6) The authors use forskolin to drive adenylyl cyclase and measure AAR genes in astrocytes when cultured with neurons. Can forskolin alone regulate AAR gene expression in astrocyte only cultures?

Yes, forskolin can also drive AAR gene expression in astrocyte mono-cultures. Our working hypothesis is that, while neuronal co-culture and contact-dependent signaling (e.g. via notch) is required for many functional changes to occur in astrocytes, the signaling machinery required to mediate AAR gene induction is intrinsic to all astrocytes. These data are shown now in Fig. S6c.

Reviewer #2 (Remarks to the Author):

In this manuscript the authors co-cultured mouse cortical astrocytes with rat cortical neurons and performed RNASeq on the cultured cells. They separated mouse from rat transcripts and identified several neuron-induced genes in astrocytes. They also co-cultured rat neurons with human astrocytes and found that transcriptional responses in human cells mirror those of mouse astrocytes. Freshly explanted astrocytes lose neuron induced genes after 4 days in culture and addition of neurons restores expression of these genes. Using a P301S transgenic mouse that expresses mutated Tau and develops Tauopathy, the authors found that neurodegeneration triggered by this mutated neuron-specific transgene leads to a decline in the astrocytic expression of neuronally-regulated genes in vivo. Hundreds of astrocytic genes were found to be acutely regulated by glutamatergic synaptic activity via mechanisms involving cAMP/PKA-dependent CREB activation. The groups of astrocytic genes induced by neurons or neuronal activity both show age-dependent decline in humans.

Overall, the concept that neurons regulate gene expression in astrocytes is important and has significant implications for various physiological and pathological conditions. The manuscript is well written and contains a lot of data. I have some concerns that affect the impact of the manuscript as follows:

We were pleased that the reviewer felt that the concept at the heart of our study is important and has significant implications. We feel that this concept of neuronally-regulated astrocytic gene expression, plus our demonstration of functional consequences for astrocytes would be of considerable interest to neuroscientists. In characterizing programs of astrocytic gene expression induced by neurons and neuronal activity, identifying mechanisms (notch signaling, CREB activation) and functional consequences (glutamate uptake, glucose metabolism) we feel this adds to our view of how inter-cell type signaling can control functional relationships within the neuro-glial unit.

As well as responding to the reviewers points we also show new data that CREB-dependent gene expression is sufficient to boost astrocytic glycolytic glucose metabolism and lactate export (Fig. S7b-f), thus complementing our existing data which showed it to be necessary for activity-dependent induction of astrocytic glycolytic glucose metabolism.

We are happy to have the opportunity to address the reviewers concerns below.

1-First, the rationale for using mixed species is not really convincing. Why not instead use transgenic animals that express a fluorescent protein in astrocytes and a different fluorescent protein in neurons and separate these cells to very high purity by flow?.

Although experienced in physical separation techniques such as FACS and immunopanning (e.g. see this study and ⁴) we decided to use the mixed species approach because with physical sorting it is extremely difficult to eliminate two major sources of error: the induction of expression artefacts, and sample purity/contamination.

Physical sorting processes involve multiple stresses including protease digestion of extracellular proteins, dramatic changes in cellular morphology, and mechanical forces and manipulations that can all trigger transcriptional responses resulting in expression artefacts. For example, in a recent (and excellent) study in *Neuron* on "Development of a Method for the Purification and Culture of Rodent Astrocytes" ⁵, gene expression in astrocytes acutely sorted from the P7 and P0 brain were compared to those after a further 7 days in culture,. For both P7 and P0 brains the most strongly down-regulated gene (microarray probeset 1373759_at) was immediate early gene *Fosb*. Furthermore, the gene lists contain many known immediate early genes such as *Nr4a1*, *Nr4a2*, and *Ier2*, also down-regulated. This means that either astrocytes have more *Fosb* and other IEGs in vivo than in vitro (for which there is no evidence), or that the sorting process

acutely induces IEG expression. This ambiguity is an issue with all physical sorting methods. To illustrate the issues in using physical sorting techniques, we applied an immuno-panning 'sorting' process to a homogeneous population of cultured astrocytes. The simple trypsinization of the cultures to generate a single cell suspension suitable for sorting was sufficient for induction of immediate early gene expression, (including *Fos*, *Fosb*, *Nr4a1*, *Nr4a2*, and *Ier2*, Fig. S2b) and the immuno-purification process modifies these genes also (Fig. S2c). A simulated FACS protocol where cells were passed through the FACS machine also induced IEG expression (Fig. S2d). These kind of gene expression artifacts are unacceptable to us, particularly when we are trying to decipher signals mediating dynamic transcriptional responses in astrocytes due to neuronal activity, and which could both add to and/or synergies with the sorting-induced signals in an unpredictable way. These issues are now articulated to better justify the approach used).

In addition to the induction of expression artifacts, the issue of sample purity/contamination is always present in physical sorting techniques. As outlined in an informative meta-analysis by Nelson and coworkers⁶, when sorting CNS cell types from each other a degree of contamination is apparent regardless of the technique (FACS, immunopanning, laser-capture microdissection). This is particularly pertinent in our case where we are trying to profile the astrocytic transcriptome in the absence or presence of neurons. In the former case, we know that the cell population is highly pure and not subject to contamination: because the astrocytes were passaged prior to final plate down, re-adhesion and survival of poorly adherent immune cells is impaired, and the cells are >99% *Aldh1h1* positive (Fig. S2a), >98% GFAP positive⁷, confirming their astrocytic identity. However, in the latter case of mixed neuronal/astrocyte populations, a confounding degree of neuronal contamination is extremely difficult to totally eliminate, regardless of the approach used⁶. The new data where we show the dramatic and widespread influence of a 5% neuronal contamination of astrocytic RNA (Fig. S3a, see also response to a later point by the reviewer) further underlines this point.

Collectively, these issues led us to develop the mixed-species approach described as a powerful tool to use, and is coupled with further validation in single species preparations (see below).

Indeed, the use of mixed species carries several hazards by itself. A major problem is that rat cortical neurons could cause changes in mouse astrocytes different that changes induced by mouse neurons because of antigenic differences and the likely presence of small numbers of contaminating immune cells such as microglia and blood cells. In this regard the system used in this manuscript does not account for such potential problems.

Regarding rat vs. mouse antigenic differences, we had tackled this question in the context of the control of mouse astrocytic gene expression (AAR-genes) induced by rat neuronal synaptic activity." *To further validate the mixed species approach we wanted to rule out the possibility that coculturing neurons and astrocytes of different species results in erroneous effects of synaptic activity on astrocytic gene expression...*". All genes studied were induced by mouse neuronal activity (old Fig. S5a, new Fig. S6d).

In the case of neuronally regulated astrocytic genes (NR-genes), the uniform decline of all the NR-genes in Fig. 3a in mouse astrocytes upon their removal from their normal in vivo environment supports the view that NR-genes identified in mouse astrocytes by assessing the influence of rat neurons on their expression, are also induced in vivo by extrinsic mouse signals (e.g. neurons). We have now gone further and analysed expression of the NR-genes shown in Fig. 3a and 3b whose induction could be tracked in a single species co-culture by virtue of its expression being >10 fold lower in neurons than even mono-cultured astrocytes (namely *Hes5*, *Dio2*, *Slco1c1*, *Glul* and *Cldn10*) and found that this group of genes is indeed induced (new Fig. S4a).

Thus, we find no evidence that the mixed species approach has a qualitative influence on the group of astrocytic genes induced by neurons or neuronal activity. Furthermore, we have now confirmed that the functional impact of neurons in inducing mouse astrocytic glutamate transporter capacity is observed for in a single species mouse neuron/astrocyte co-culture (Fig. S4b) as well as the original mixed species preparation

Additionally, we have further evidence that the ability of neurons and astrocytes to send

and receive signals respectively to modulate astrocytic gene expression is evolutionarily well-conserved. In the context of human cells, we had already shown that rat neurons can induce morphological transformation of human astrocytes (Fig. 2c,d), as well as induce Notch target *Hes5/Hey2* and glutamate transporter genes *Slc1a2* and *Slc1a3* (Fig. 2e and Fig. 4d), suggesting that human astrocytes are similarly receptive to maturation signals derived from rat neurons as mouse astrocytes are. Moreover, we now show that in a 'reverse' co-culture of mouse neurons with rat astrocytes, that the mouse neurons induce Notch targets *Hes5/Hey2*, as well as glutamate transporter genes *Slc1a2* and *Slc1a3* (Fig. S5c and see response to Reviewer 3).

This conservation perhaps not surprising given that in the context of NR genes, a key putative signal is Notch, a highly conserved pathway. For example, the 1400 amino acid stretch of EGF repeats in Notch2 responsible for binding notch ligands is >97% conserved mouse vs. rat in the major astrocytic notch, and major neuronal notch ligand Jagged 2 is also >97% conserved mouse vs. rat. Conservation of these regions mouse vs. human is also high-around 95%. Nevertheless, despite the above, we do now acknowledge it as a possibility that we cannot formally rule out, but feel that its advantages outweigh its disadvantages for answering our research questions.

Regarding the purity of cell populations: as noted above, the astrocytes were found to be >99.5% Aldh1h1 positive. Thus, we are confident that we are reporting gene expression changes induced specifically in astrocytes. As for minor numbers of immune or endothelial cells in the neuronal preparation overlaid onto the astrocytes, this is not an issue that is specific to a mixed species preparation, but any experiment of this type. In studying the effects of TTX-sensitive neuronal action potential burst activity on astrocytic gene expression (AAR genes), it is extremely hard to envisage how a non neuronal cell can be inducing the effects observed, given the protocol used to induce neuronal burst activity (TTX washout). In the context of the 'NR gene' induction: could tiny numbers of immune or endothelial cells be important mediators of the notch-dependent maturation signals affecting astrocyte gene expression and glutamate uptake capacity? To us this seems unlikely, not least because notch is a cell contact-dependent juxtacrine signal, and therefore is unable to act over long distances on many cells, as would be required of a hypothetical signal emanating from a minority contaminating cell type.

In the co-culture system employed in the manuscript, the rat neuronal preparation is co-cultured onto the mouse astrocytic lawn in zero serum and no other added mitogens to limit proliferation of non-neuronal cells. To limit this even further we have performed an experiment with an adapted protocol such that the rat neuronal preparation is made and added to the astrocytes in the continuous presence of anti-mitotic agent AraC. This prevents the proliferation of non-neuronal cells, although the pre-existing lawn of mouse astrocytes remains. Under these conditions, >99% of the rat neuronal preparation is Neurochrome+ neurons (Fig. S5d), <0.01% Aldh1h1+ astrocytes, <0.01% Iba1+ microglia, after 9 days in culture. The only caveat to these experiments performed in the presence of AraC is that the astrocytes do not appear as healthy as in standard zero serum conditions: there is a small but perceptible level of cell death and upon patching the astrocytes their resting membrane potential was found to be slightly more depolarized than normal. Nevertheless, the group of putative notch-dependent neuronally-regulated astrocytic genes studied in Fig. 4c are still induced (Fig. S5e), consistent with the notion that neurons are the key mediators of these changes.

We feel these new data will address the reviewer's concerns, but nevertheless acknowledge in the manuscript the formal possibility that non neuronal cells could be contributing to the observed transcriptional changes.

2-Most of the data generated in this manuscript is based on cells in culture. It is well established that cells in culture change their gene expression profiles and are different from cells in vivo. This puts into question the relevance of the data shown in the manuscript. Even experiments showing the effects of neuronal activation, can now be done in vivo (by optogenetics for example).

The first part of the manuscript is centred on the first genome-wide profiling of the influence of neurons on astrocytic gene expression. It is unclear how this question can be tackled solely in vivo, since astrocytes are always in the presence of neurons, so the role of neurons in shaping their gene expression and functional properties would be extremely hard to address. However, having identified the transcriptional changes induced by neurons in astrocytes in vitro, we devote

significant space to addressing the relevance of these transcriptional changes to the *in vivo* state. For example, we show that the influence of neurons on the astrocytes pushes its transcriptome towards a more *in vivo*-like profile (Fig. 2a,b), and that the neuronal influence on expression of astrocytic genes (up- or down) aligns significantly with the developmental trajectory of those same genes in the human brain (Fig. 2f,g). We also show that expression of neuronally-induced astrocytic genes decline if the astrocytes were removed from their normal (neuron-containing) *in vivo* environment, and that the subsequent addition of neurons *in vitro* reverses this loss of expression (Fig. 3a,b). These observations are consistent with a neuronally-derived signal being responsible for maintaining expression *in vivo* of the genes we identified *in vitro*. In further support of this, we show that neuronally-induced astrocytic gene expression declines *in vivo* in a model of neurodegenerative tauopathy (Fig. 3e). Further *in vivo* relevance of this group of neuronally-induced astrocytic genes is shown in the context of the ageing human brain, where expression of this group of genes declines in astrocytes from humans over the age of 40, compared to those under 40 (Fig. 7h).

The second part of the manuscript is centred on uncovering the concept that synaptic activity can influence gene expression in astrocytes. We feel that it is essential to elucidate these transcriptional changes, and the underlying mechanisms, by taking a reductionist approach. While manipulating neuronal activity *in vivo* by optogenetics is possible, the profile of activity required to induce gene expression is unclear. Moreover, the physical sorting required to isolate the astrocytes would induce expression artifacts (as discussed above). It is worth noting that in the context of activity-dependent *neuronal* gene expression, genes identified as being controlled by neuronal activity *in vitro*, such as *Fos*, *Npas4*, *Arc*, *Bdnf* and *Txnip*, were all subsequently confirmed to be controlled by neuronal activity *in vivo*. Thus, the differences in activity-inducibility *in vitro* vs. *in vivo* may be quite modest. Indeed, we show that sedating doses of dizocilpine reduces expression of activity-regulated astrocytic genes (Fig. 5g) and that intriguingly, expression of activity-regulated astrocytic genes declines in astrocytes from humans over the age of 40, compared to those under 40 (Fig. 7i).

Over the next 2-3 years our plans are to delve more deeply into astrocytic processes *in vivo*, and their capacity for non cell-autonomous neuroprotection, by exploiting the mechanistic insight we have gained from the current study.. However, the timescale of these experiments rules them out for the current study, which we feel is already extensive in its breadth and depth of data. We also note that in our decision letter, the editors of *Nature Communications* stated that they "*would be prepared to overrule Reviewer #2's pt 2 regarding the use of in vitro over in vivo data*".

3-The experiment described on page 5 To assess the usefulness of the mixed species approach in avoiding cell-type contamination associated with physical separation approaches, by simulating a sorting process that achieved 95% purity, and 'contaminating' the astrocytic mouse reads with the sorted rat (primarily neuronal) reads such that they formed 5% of total reads is not appropriately done. Instead, the authors should actually mix the actual cells in culture to contain an 95% to 5% ratio then perform the RNASeq and see if that affects the results.

The reviewer does not explain why they feel that this simulation is "not appropriately done" which makes it hard to address this point. One possibility is that they would prefer that the experiment was actually done, rather than simulated, in case the relative amounts of mRNA in a neuron and an astrocyte are different (the simulation assumes that amounts are the same in both cells). The reviewer's suggestion to mix the actual cells in culture to contain an 95% to 5% ratio then perform the RNA-seq would not, in our opinion, be a useful experiment. The reason for this is that not only would the effects of neuronal contamination be apparent, but so would the influence of those neurons on the astrocytic transcriptome, which would therefore confound the interpretation of the results. We have taken a different approach to get round this: neurons and astrocytes were cultured separately and subjected to a cell count to assess density. RNA was then extracted from both neurons and astrocytes, and neuronal RNA added to the astrocytic RNA at an amount corresponding to a 5% neuronal contamination *by cell number*. RNA-seq was then performed on the 'contaminated' and 'pure' astrocytic samples and differential gene expression analysis performed. The effects of just 5% cell number contamination were very substantial: 863 genes were expressed >2-fold higher in the 95% sample ($P_{adj} < 0.05$), compared to the pure astrocytic sample (Fig. S3a, Supplemental Table 7). Of these 863 genes, 216 were expressed >10-fold

higher, i.e. highly neuron-enriched genes such as *Syp* (synaptophysin), *Nefl* (neurofilament light) and *Syt4* (synaptotagmin IV). Extrapolating from these numbers: if 5% contamination enriches a gene by 10-fold, a 1% contamination would be expected to enrich the same gene by 2.8-fold. Thus, even sorting to 99% purity would result in very large numbers of false positive differential gene expression hits when trying to understand the influence of neurons on astrocytic gene expression. This underlines the usefulness of *in silico* read separation.

4-Cortical and spinal cord astrocytes are different. Therefore it is not appropriate to use spinal spinal cord cells in the experiments described in figure 3. Cortical cells should be used instead.

The reason we studied the spinal cord is that this is the area of the CNS where neurodegeneration is most consistent in the P301S tauopathy mouse. However, while we show that expression of neuronally-regulated astrocytic genes declines in the P301S spinal cord (Fig. 3e), the reviewer is correct that our definition of "neuronally regulated astrocytic genes" is based on our observations on cortical astrocytes, not spinal cord astrocytes. We have now performed experiments on spinal cord astrocytes, cultured in the presence or absence of neurons, and found that the group of genes analysed in Fig 3e are indeed induced in spinal cord astrocytes by neurons (Fig. S4c), meaning that our prior 'assumption' is now grounded in experimental observation.

Reviewer #3 (Remarks to the Author):

This study presents interesting and original data about the influence of neurons and neuronal activity on the regulation of gene expression in astrocytes using RNA-Seq. The authors overcome the issues of cell-type contamination and artifacts due cellular stress during physical separation of cells by establishing an astute in silico sorting procedure with co-cultures of neurons and astrocytes from different species. In addition, convincing control experiments were carried out to validate their mixed species approach during the different experimental tests (mature and human astrocytes, and the influence of synaptic activity).

This is an extensive piece of work and a well written manuscript which conveys important messages about neuron-to-astrocyte communication and the role of neuronal activity in shaping the functional properties of astrocytes such as neurotransmitter uptake and metabolic homeostasis. In addition, specific signaling mechanism were identified in each case and validated with physiological approaches.

This paper should be of great interest to the reader

We were very pleased that the reviewer felt our study to be interesting and original, conveying important messages about neuron-to-astrocyte communication. We address their points below.

The presentation of the results, however, is too narrative and could benefit from more quantitative descriptions. It would be instructive to have access to experimental parameters such as read counts, percentages, cutoffs, etc, and to how genes were identified and regulated in each experiment without having to check each table. Perhaps a summary table could be embedded in the results section displaying these information.

We have been through the manuscript to include more quantitative descriptions, and articulation of experimental parameters without resorting to supplemental information. In some cases this includes transferring this information into the main text, and in other cases we felt its incorporation into the legends was more appropriate. We have also added the information specifying the exact numbers of read counts (attributable to each species), FPKM cut-offs, differentially expressed genes, and other information for the RNA-seq data in Fig. 1 and Fig. 5.

The statistical analyses (or their descriptions) also need to be extended. For example, it isn't clear whether data from the RNA-seq analysis are coming from one sample or whether they have been repeated using replicates from the same or independent cultures.

All gene expression analyses, including the RNA-seq comes from independent cultures of distinct tissue origin (i.e. they are biological replicates, not technical replicates). No culture, be it neuronal

or astrocytic, ever contributes more than 1 data point to any statistical analysis. This is now made clearer in the methods.

Likewise, it is unclear how the statistical significance is established when comparing gene sets for NR- or AAR-genes to gene sets published in previous paper. Proper statistical test should be used and described.

We felt that we described the process reasonably thoroughly in the figure legend. For example for Fig. 2a we state:

" Gene fold-change caused by neuronal co-culture (as analysed in Fig. 1c) is shown for those genes identified by Cahoy et al. (2008) in a microarray screen as being either elevated (A) or lowered (B) in astrocytes in vivo, compared to in vitro mono-culture¹. Genes expressed >0.5 FPKM in either mono- or co-culture are shown. In some cases gene names are updated from those quoted in Cahoy et al. (2008). Within the group of genes elevated in vivo (A, 695 genes induced >2-fold) the influence of neurons on astrocytic expression is shown, ranked according to fold-change. P=2.5E-6 (paired t-test comparing FPKM of these 695 genes in mono-culture vs. co-culture)."

In other words, while the figure only shows the fold-change for each of the genes (co-culture vs. mono-culture), the statistical test involves a paired t-test for all 695 genes (co-culture FPKM vs. mono-culture FPKM). A low p-value indicates a significant trend for these genes to be altered (in co-culture vs. mono-culture) in a particular direction (in this case, up). To further clarify the process, we now include, for all figures where similar comparisons are made, supplemental tables that lists the genes, the fold-change that defines their inclusion (i.e. the criteria stated on the x-axis), the fold-induction (y-axis) and the related FPKM values that are subject to a paired t-test leading to the p-value stated on the graph. Data relating to the comparisons performed in Fig. 2a, 2b, 2f, 2g, 7h and 7i are presented in Supplemental Tables 9, 10, 17, 18, 15 and 16 respectively.

Minor comments:

In the assessment of the species-specific sorting RNA-Seq workflow, the authors evaluate the attribution of reads to species-specific genomes using only mouse RNA-seq data sets. What would have been the outcome of these experiments with data sets from the rat instead ?

The rat reads are overwhelmingly neuronal, as would be expected. One question under investigation in the laboratory at present is the influence of astrocytes on the neuronal transcriptome (i.e. the reciprocal influence to the topic of this study). This could not be answered with the current data set because we did not have samples that comprised rat neurons in the absence of astrocytes.

Along this line, would the findings described in the subsequent experiments be the same if the experiments were conducted with mouse instead of rat neurons plated on top of rat instead of mouse astrocytes ?

This is a very good suggestion. We have performed this experiment, swapping the species of neurons and astrocytes (i.e. Astr_{Rat}- Neur_{Mus} co-cultures, instead of Astr_{Mus}- Neur_{Rat} co-cultures). We looked at expression of notch target genes *Hes5* and *Hey2*, as well as glutamate transporter genes *Slc1a2* and *Slc1a3*, using rat-specific qPCR primers (Fig. S5c). We observed that mouse neurons induced these genes in rat astrocytes, just as rat neurons induce them in mouse astrocytes, further supporting the generality of the mechanisms described.

The expression threshold (in FPKM) to exclude genes from analyses are not the same across all experiments. Why is that ? How would it influence the outcome of each analysis ?

There is no specific reason for this, and we thank the reviewer for bringing it to our attention. For greater consistency we have harmonized all cut-offs in all analyses to >0.5 FPKM across the conditions compared. No conclusions have altered as a result of the slightly adjusted gene lists and analyses. Because different numbers of genes were included in the analyses in Fig. 2f, 2g, 7h and 7i as a result of adjusting the FPKM threshold, with the p-values changed slightly from 0.002 to 0.0016 (2f), 0.007 to 0.003 (2g), 0.0007 to 0.0002 (7h) , and 0.003 to 0.004 (7i) respectively.

(Note that in the original manuscript, the cut-off for Fig. 1c was already >0.5 FPKM, not >2 FPKM as stated in the legend (apologies for this error).

For astrocytic genes induced by synaptic activity the cutoff was >1.3 whereas it was >2-fold for most of the other experiments of the paper. Could the author explain the relevance of this change?

For astrocytic genes induced by synaptic activity (AAR-genes) we highlighted genes induced >1.3-fold in the scatter graphs (Fig. 5), and for neuronally-regulated genes (NR-genes) we highlighted genes induced >1.5-fold in the scatter graph (Fig. 1). These thresholds were fairly arbitrary and in hindsight should have been the same as each other. However, the reviewer is correct that we used a >2-fold cut-off for our comparative gene list analyses throughout the manuscript, where the intention is to compare the strongest induced genes. On reflection it makes better sense to highlight the genes up- or down-regulated >2-fold on the scatter plots in Figs 1 and 5, since it is these genes that feed into the comparative gene list analyses. This has now been rectified. For both NR- and AAR-genes we also highlight genes more modestly (but significantly: $P_{\text{adj}} < 0.05$) regulated between 1.3- and 2-fold, using a different colour.

Samples label “young” (age 8-35) compared to “older” (47-63) in figure 7 are in conflict with the terminology “mature” (age 8-63) compared to “fetal” (18-19 gestational weeks). This is confusing. I suggest that the authors choose more harmonious descriptive terms.

Rather than "mature" (age 8-63) we now use the term "post-natal".

There is already a lot of material with the various comparisons done in this manuscript. In the discussion, pointing out all the advantage of mixed species RNA-seq is pertinent but I don't see the point of commenting experiments which are not described in this paper. This should be removed.

We agree and have removed this section alluding to specific future plans and restricted our description of the advantages and potential uses of the technique to general principles and applications.

References

1. Cahoy JD, Emery B, Kaushal A, Foo LC, Zamanian JL, Christopherson KS, Xing Y, Lubischer JL, Krieg PA, Krupenko SA, Thompson WJ, Barres BA. A transcriptome database for astrocytes, neurons, and oligodendrocytes: a new resource for understanding brain development and function. *J Neurosci* 2008, **28**(1): 264-278.
2. Zhang Y, Sloan SA, Clarke LE, Caneda C, Plaza CA, Blumenthal PD, Vogel H, Steinberg GK, Edwards MS, Li G, Duncan JA, 3rd, Cheshier SH, Shuer LM, Chang EF, Grant GA, Gephart MG, Barres BA. Purification and Characterization of Progenitor and Mature Human Astrocytes Reveals Transcriptional and Functional Differences with Mouse. *Neuron* 2016, **89**(1): 37-53.
3. Zamanian JL, Xu L, Foo LC, Nouri N, Zhou L, Giffard RG, Barres BA. Genomic analysis of reactive astrogliosis. *J Neurosci* 2012, **32**(18): 6391-6410.
4. Bell KF, Al-Mubarak B, Martel MA, McKay S, Wheelan N, Hasel P, Markus NM, Baxter P, Deighton RF, Serio A, Bilican B, Chowdhry S, Meakin PJ, Ashford ML, Wyllie DJ, Scannevin RH, Chandran S, Hayes JD, Hardingham GE. Neuronal development is promoted by weakened intrinsic antioxidant defences due to epigenetic repression of Nrf2. *Nat Commun* 2015, **6**: 7066.
5. Foo LC, Allen NJ, Bushong EA, Ventura PB, Chung WS, Zhou L, Cahoy JD, Daneman R, Zong H, Ellisman MH, Barres BA. Development of a method for the purification and culture of rodent astrocytes. *Neuron* 2011, **71**(5): 799-811.

6. Okaty BW, Sugino K, Nelson SB. A quantitative comparison of cell-type-specific microarray gene expression profiling methods in the mouse brain. *PLoS ONE* 2011, **6**(1): e16493.
7. Soriano FX, Leveille F, Papadia S, Higgins LG, Varley J, Baxter P, Hayes JD, Hardingham GE. Induction of sulfiredoxin expression and reduction of peroxiredoxin hyperoxidation by the neuroprotective Nrf2 activator 3H-1,2-dithiole-3-thione. *J Neurochem* 2008, **107**(2): 533-543.

REVIEWERS' COMMENTS:

Reviewer #1 (Remarks to the Author):

The authors addressed my concerns in their revision.

Reviewer #2 (Remarks to the Author):

The authors adequately addressed part of my concerns. It remains unclear if the data presented in the manuscript are relevant *in vivo*, however, I do not think this weakness should prevent the manuscript from being published because the concept presented is novel and may have important implications.

Reviewer #3 (Remarks to the Author):

The authors have satisfactorily addressed the points that I raised. Furthermore by also addressing those of the other reviewers the manuscript is considerably improved.